# PACE: PRETRAINED AUDIO CONTINUAL LEARNING

**Chang Li**,* **Kanglei Zhou**,* **Liyuan Wang**[†]
Department of Psychological and Cognitive Sciences, Tsinghua University

## ABSTRACT

Audio is a fundamental modality for analyzing speech, music, and environmental sounds. While pretrained audio models have significantly advanced audio understanding, they remain fragile in real-world scenarios where data distributions evolve over time. In this work, we present the first systematic benchmark for audio continual learning (CL) with pretrained models (PTMs) and provide a comprehensive analysis of its unique challenges. Unlike in the vision domain where parameter-efficient fine-tuning (PEFT) has proven effective for CL, directly applying such strategies to audio leads to poor performance. This is due to a fundamental property of audio backbones: they emphasize low-level spectral details rather than structured semantics, resulting in severe upstream–downstream misalignment. Through extensive empirical analysis, we identify a promising technical route based on analytic classifiers with first-session adaptation (FSA), but also uncover two major limitations: representation saturation in coarse-grained scenarios and representation shifts in fine-grained scenarios. To address these challenges, we propose **PACE**, an innovative method that improves FSA via a regularized analytic classifier and introduces multi-session adaptation through adaptive subspace-orthogonal PEFT for better semantic alignment. Additionally, we design spectrogram-based boundary-aware perturbations to mitigate representation overlap and improve stability. Experiments across six diverse audio CL benchmarks demonstrate that PACE substantially outperforms state-of-the-art baselines, representing a significant step toward robust and scalable audio CL with PTMs.

## 1 INTRODUCTION

Audio is central to human communication and environmental perception, supporting numerous applications such as speech recognition (Abdel-Hamid et al., 2014), acoustic event detection (Zhuang et al., 2010), and sound scene understanding (Nakamura et al., 2000; Zhou et al., 2024c). With the rise of large-scale supervised (Gong et al., 2021) and self-supervised pretraining (Gong et al., 2022; Chen et al., 2024; 2022; Li et al., 2024b), pretrained audio models have achieved remarkable success across a wide range of downstream tasks. However, in real-world scenarios where audio distributions evolve continuously, these models often struggle to effectively adapt without incurring catastrophic forgetting, exposing a key limitation for audio-related applications.

Continual learning (CL) aims to address this limitation by enabling models to learn new tasks while retaining old knowledge (Zhou et al., 2025b; 2024d; 2025a). While recent progress in the vision domain has demonstrated the effectiveness of parameter-efficient fine-tuning (PEFT) for CL with pretrained models (PTMs) (Wang et al., 2022b;a), their extension to audio remains largely underexplored and highly non-trivial. Pretrained vision models generally encode stable and well-structured semantic representations (Janson et al.),

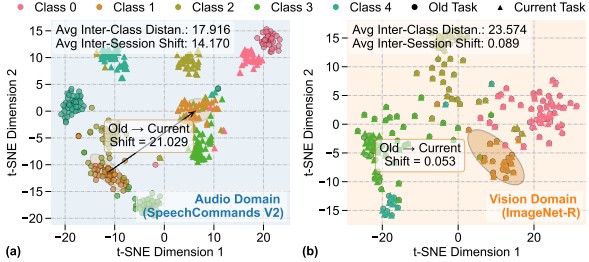

Figure 1: Audio CL (a) on SC2 suffers from clearly much stronger **representation shifts** between adjacent sessions than vision CL (b) on ImageNet-R.

---

*Co-first author
[†]Corresponding author: liyuanwang@tsinghua.edu.cn

leading to relatively mild representation shifts across adjacent sessions (see Fig. 1(b)). In contrast, audio recognition depends heavily on fine-grained spectral cues (Wang et al.; Wu et al.), whereas CL requires progressively adapting high-level semantic objectives to acquire discriminative representations. Pretrained audio models (Chen et al., 2024; Gong et al., 2022; Alex et al., 2025), typically trained via spectrogram reconstruction objectives, prioritize low-level time-frequency patterns over structured semantics (Tabassum et al., 2024; Bao et al.; Epstein & Meir, 2019). Despite the learning objective, the **upstream–downstream mismatch** can be further enlarged by discrepancies between the pretraining and downstream data distributions, reflected in Fig. 10. These mismatches force the backbone to continually reshape its internal representations across sessions, inducing substantial representation shifts that often surpass the subtle spectral differences between classes, leading to severe catastrophic forgetting (see Fig. 1(a)).

To systematically investigate whether and how pretraining-based CL methods can be effectively applied to audio, we construct a comprehensive benchmark and uncover **three key findings**: **First**, we find that representative vision-domain CL methods, particularly those relying on task-shared representations, exhibit significant performance degradation when transferred to audio. We identify a simple but effective technical route, which integrates first-session adaptation (FSA), backbone freezing, and second-order analytic classification (McDonnell et al., 2023; Zhuang et al., 2022). **Second**, although this approach achieves strong performance on coarse-grained benchmarks with relatively small domain gaps, it exhibits representation saturation when learning the first task (Li et al., 2020), which hinders subsequent adaptation. **Third**, this approach becomes less effective when applied to the more demanding fine-grained scenarios (e.g., musical instrument and speaker classification) that involve substantial upstream-downstream mismatch. As a result, a pronounced performance gap remains compared to the joint training upper bound.

To close this gap, we propose PACE (**P**retrained **A**udio **C**ontinual l**E**arning), a novel method designed to fully harness pretrained audio models while overcoming upstream-downstream mismatch in both coarse- and fine-grained CL scenarios. Unlike vision CL where freezing the pretrained backbone often suffices (Zhang et al., 2023; 2024), PACE selectively adapts the later backbone layers with an audio-specific PEFT strategy tailored for FSA, enabling more effective representation learning, particularly on coarse tasks. To extend adaptability across sessions in fine-grained scenarios, PACE further introduces (1) multi-session adaptation (MSA), which incorporates an adaptive subspace-orthogonal PEFT strategy to enable progressive adaptation while constraining the drift of previously learned features, thereby achieving a principled balance between stability and plasticity; and (2) a boundary-aware perturbation mechanism, which applies targeted time–frequency transformations to approximate historical decision boundaries, enhancing intra-class compactness and inter-class separability in the learned representation space.

We conduct extensive experiments across three coarse-grained benchmarks (ESC-50, US8K, SC2) and three fine-grained benchmarks (TIMIT-2, TIMIT-3, VocalSet). PACE consistently outperforms state-of-the-art CL methods, with notable gains of at least +5.3% on TIMIT-2, +4.1% on TIMIT-3, and +6.3% on VocalSet. Moreover, it significantly reduces the gap to the joint training upper bound, achieving performance within 0.8% on ESC-50, 0.6% on US8K, 3.5% on SC2, 4.3% on TIMIT-2, 1.2% on TIMIT-3, and 7.6% on VocalSet. To facilitate future research, we will release all constructed benchmarks and reproduced baselines along with our codebase.

## 2 BENCHMARKING AUDIO CONTINUAL LEARNING

To systematically investigate audio CL with PTMs, we first introduce the problem formulation and then present comprehensive benchmark results that reveal the unique challenges of this setting.

**Pretrained CL.** CL with PTMs assumes access to a pretrained backbone $f_0$ parameterized by $\theta_0$ obtained from a source domain, which is incrementally adapted to a sequence of $T$ tasks $\mathcal{T}_1, \cdots, \mathcal{T}_T$ without retraining from scratch. Each task $\mathcal{T}_t = (\mathcal{D}_t, \mathcal{Y}_t)$ updates the model from $\theta_{t-1}$ to $\theta_t$ using $\mathcal{D}_t$, and evaluation is performed over the accumulated label space $\bigcup_{i=1}^{t} \mathcal{Y}_i$.

**Pretrained Audio CL.** In audio CL, each input $x_{n,t} \in \mathcal{X}_t$ is a raw audio signal, and each label $y_{n,t} \in \mathcal{Y}_t$ belongs to a task-specific category, where $\mathcal{Y}_i \cap \mathcal{Y}_j = \emptyset$ for $i \neq j$. The objective is to learn from sequential datasets $\mathcal{D}_1, \cdots, \mathcal{D}_T$ while preserving performance on all previous classes. Unlike

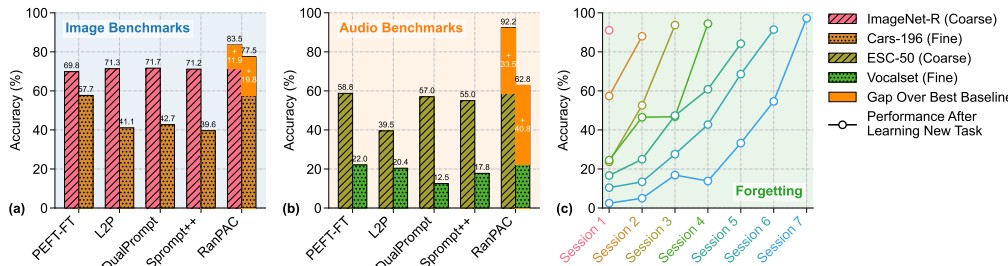

Figure 2: Comparison of vision CL and audio CL. (a) and (b) present performance patterns on audio and image datasets in both coarse- and fine-grained settings. (c) shows that, despite strong first-task plasticity with PEFT-FT, large representation shifts lead to severe forgetting.

vision CL, pretrained audio models face additional challenges in this setting due to a fundamental mismatch between pretraining objectives and downstream task granularity.

In our empirical evaluation, we adopt EAT (Chen et al., 2024), a general-purpose audio self-supervised model pretrained on large-scale audio and speech datasets (Gemmeke et al., 2017), as the default $f_0$. We construct six representative audio CL benchmarks. The first three, ESC-50 (Piczak, 2015), UrbanSound8K (Salamon et al., 2014), and Speech Commands V2 (SC2) (Warden, 2018), represent **coarse-grained** tasks such as environmental sound classification and keyword spotting. These tasks are relatively well aligned with the EAT pretraining objective (Li & Angelov, 2024), leading to a comparably smaller upstream–downstream mismatch. To explore more challenging scenarios involving severe distribution changes, we further introduce three **fine-grained** benchmarks: TIMIT-2&3 (Garofolo et al., 1993) for speaker identification, and VocalSet (Wilkins et al., 2018) for musical instrument recognition. These tasks demand structured semantic understanding that is notably misaligned with EAT's pretraining, thus posing greater challenges for CL. Detailed dataset description and task configurations are provided in Sec. 4.1 and Sec. D.

From the benchmarking results, we identify three key empirical findings:

**Finding 1: Vision CL methods degrade on audio tasks.** As shown in Figs. 2(a) and 2(b), directly transferring CL methods from the vision domain to audio understanding tasks yields different performance patterns. In particular, PEFT-based CL methods such as L2P (Wang et al., 2022c), DualPrompt (Wang et al., 2022b), and S-Prompt++ (Wang et al., 2022a) exhibit significantly worse performance in the audio domain, with degradation nearly three times larger than in vision. These methods rely on shared representations for prompt-key matching or task-incremental adaptation, which appear less effective when handling the fine-grained spectral structures of audio. In contrast, statistics-based methods use a once-tuned backbone with an analytic classifier built on second-order statistics, exemplified by RanPAC (McDonnell et al., 2023) and ACL (Zhuang et al., 2022), and consistently deliver stronger, more stable results compared to PEFT-based methods in audio CL. We attribute this performance gap to the pronounced representation shifts in the audio domain (see Figs. 1(a) and 1(b)), which manifest in rapid and substantial forgetting once a new session is learned (see Fig. 2(c)). **While RanPAC sacrifices model plasticity for continual updates, it mitigates the more severe audio representation shifts, making this trade-off more suitable for audio CL.** These observations motivate us to adopt an analytic classifier with second-order statistics upon a frozen backbone as the **foundational technical route** for pretrained audio CL.

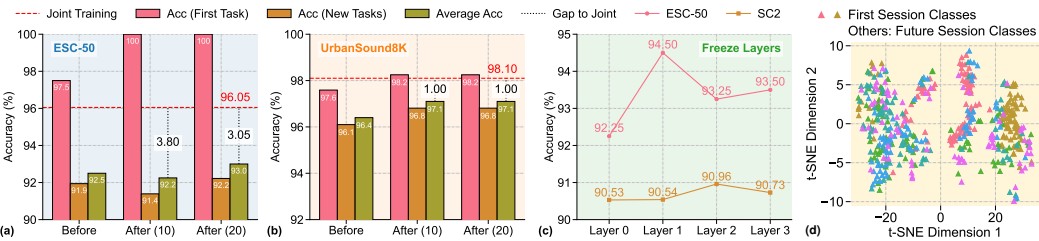

Figure 3: Analysis of representation tuning in CL. (a) and (b) show RanPAC's first-session, future-session, and average performance across FSA epochs relative to joint training. (c) shows the gains from simply freezing shallow layers. (d) is the t-SNE visualization of VocalSet after FSA.

**Finding 2: Representation saturation on coarse-grained datasets.** As shown in Figs. 3(a) and 3(b), RanPAC achieves high first-session accuracy on coarse datasets even without FSA, suggesting a relatively small domain gap between pretraining and downstream tasks in these scenarios. Although FSA further improves first-session accuracy, it fails to meaningfully improve future-task accuracy, even with extended training, leaving a noticeable gap to the joint training upper bound. This indicates that the pretrained backbone already captures most of the relevant information in the first session, limiting its ability to extract additional discriminative features to benefit subsequent tasks, a phenomenon we refer to as **representation saturation**. Furthermore, Fig. 3(c) shows that freezing shallow layers during FSA improves performance, while full-layer tuning often degrades it, even falling below the no-FSA baseline. This highlights the risk of blindly fine-tuning all layers, which can erode generalizable low-level representations obtained from pretraining.

**Finding 3: Larger performance gap on fine-grained benchmarks.** When applied to fine-grained scenarios, the identified technical route (FSA and analytic classification) quickly degrades, exposing substantial limitations caused by upstream–downstream mismatch. As shown in Fig. 3(d), while FSA improves clustering of first-session classes, it fails to produce coherent distributions for future-session classes. Moreover, Table 1 shows that extended training on the first session can worsen performance on subsequent tasks, suggesting a strong tendency toward overfitting. This effect is

| Method | TIMIT-2 | VocalSet |
|---|---|---|
| w/o FSA | 75.87 | 61.51 |
| Naive FSA | 89.92 | 62.85 |
| Extended FSA | 83.25 | 61.18 |
| Joint Training | 95.22 | 76.65 |

Table 1: Preliminary results on fine-grained benchmarks.

especially pronounced in datasets with high semantic complexity. For instance, on VocalSet, the performance gap relative to joint training reaches 13.8%, compared to only 3% and 1% on ESC-50 and UrbanSound8K, respectively. These results suggest that first-session data alone is insufficient to bridge the semantic gap between pretraining and downstream objectives in fine-grained audio tasks. Together, these findings highlight the need for progressively aligning pretrained representations with downstream tasks over **multiple sessions**, while avoiding overfitting in early-stage adaptation.

## 3 OUR PACE METHOD: PRETRAINED AUDIO CONTINUAL LEARNING

### 3.1 NOTATIONS AND DESIGN OVERVIEW

**Notations.** Let $f(\cdot)$ denote a pretrained backbone with $L$ layers and $g(\cdot)$ a classification head. Given an input audio signal, we first compute its time-frequency map $x \in \mathbb{R}^{T_x \times F_x}$ (e.g., via STFT followed by Mel filtering), where $T_x$ and $F_x$ are the numbers of frames and Mel bins, respectively. The backbone produces a representation $z = f(x) \in \mathbb{R}^D$, where $D$ is the feature dimension. $z$ is then passed through the head to predict class probabilities $\hat{y} = g(z)$ in $\mathcal{Y}$.

**Design Overview.** Motivated by the empirical findings in Section 2, we introduce **PACE** (Fig. 4), a unified, stage-wise framework for realigning pretrained audio representations with continual learning (CL) objectives. We decompose the problem into two components: the **pretrained backbone** and the **output head**, and design targeted strategies for each.

**Improved First-Session Adaptation (FSA).** Empirical evidence from **Finding 1** and the embedding-space analysis in Fig. 1 shows that naive backbone adaptation causes **severe cross-session representation shift**. This motivates adopting an analytic classifier as the default technical route. As shown in **Finding 2**, pretrained audio models already encode strong coarse-grained semantics, making them prone to **first-session saturation**. Naively fine-tuning the output head distorts these representations, limiting forward learning. To address this, we propose **improved first-session adaptation (FSA)** that freezes the output head and adapts only deeper layers using LoRA modules $\{A_1^l B_1^l \mid L_{\text{tune}} < l \leq L\}$. Once adapted, the parametric head $h^1(\cdot)$ is replaced with an analytic classifier $\phi^1(\cdot)$ to preserve pretrained semantics while avoiding unnecessary parameterization.

**Multi-Session Adaptation (MSA) with Subspace Projection.** Although FSA performs well on coarse-grained tasks, it is insufficient for fine-grained scenarios where a stronger **upstream–downstream mismatch** exists. Thus, we introduce **multi-session adaptation (MSA)**, which progressively aligns representations using multiple sessions' distributions. However, as revealed by **Findings 1 & 3**, naive MSA exacerbates the **representation shift**, resulting in catastrophic forgetting. To ensure replay-free solution, we allow adaptation only when needed and constrain it

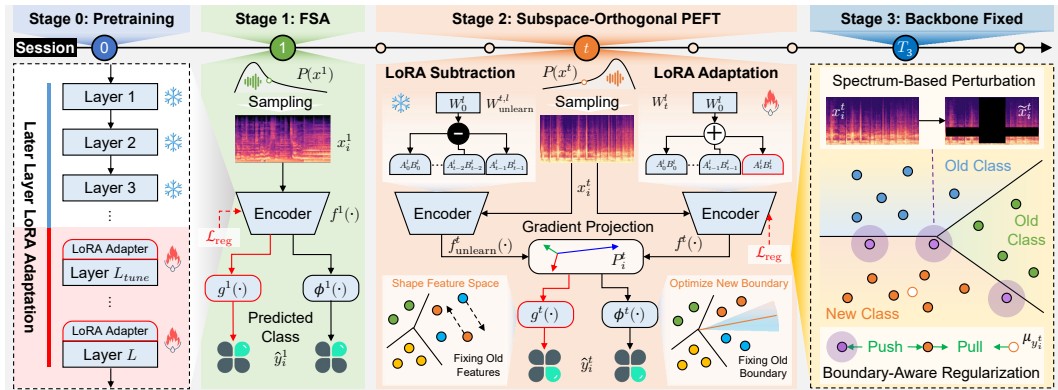

Figure 4: The proposed PACE framework. Stage 1 performs first-session adaptation with LoRA, followed by analytic inference. Stage 2 introduces subspace-orthogonal PEFT via LoRA subtraction and gradient projection. Boundary-aware regularization involves adaptation in the first two stages. Stage 3 freezes the backbone for stable adaptation. ❄: frozen; ♦: tuning; ⟶: adaptation path.

via subspace-orthogonal MSA. In sessions $2 \le t \le T_3$, gradients are projected onto an interference-free subspace $\mathcal{U}_t^l$ (via LoRA subtraction (Liu & Chang, 2025)), enabling controlled updates without distorting earlier knowledge. Inference remains analytic via $\phi_t(\cdot)$.

**Boundary-Aware Regularization.** One remaining challenge highlighted by **Finding 3** is the **overlapping class boundaries**. As representations stabilize, new classes may be forced into suboptimal, overlapping regions. This degrades both adaptation and generalization. To mitigate this, we introduce **boundary-aware regularization**. For each sample $x_{i,t}$, we generate perturbed variants $\tilde{x}_{i,t}$ that approximate class-boundary regions $\mathcal{B}t$. During training, $x_{i,t}$ is pulled toward its class prototype $\mu(x_t)$ and pushed away from $\mathcal{B}_t$, increasing inter-class margins and promoting compact, separable representations. This reduces boundary collisions and improves stability during MSA.

Each component of PACE is detailed in the following sections, including the improved FSA (Sec. 3.2), and the subspace-orthogonal MSA (Sec. 3.3) with boundary-aware regularization.

## 3.2 IMPROVED FIRST-SESSION ADAPTATION (FSA)

Empirical analysis shows that in coarse-grained audio CL, pretrained models already provide strong semantic priors. However, naively fine-tuning the full model during the first session tends to disrupt these well-aligned representations, leading to early saturation. This indicates that the first session should emphasize targeted refinement, not full backbone adaptation. To this end, our improved FSA incorporates three key ideas: (1) restricting updates to the output head so that gradients flow primarily into the backbone; (2) adapting only deeper, semantic-relevant layers; and (3) replacing the trainable head with an analytic classifier after adaptation to ensure stability in later sessions.

**Restricted Head Learning.** Existing FSA methods jointly train a linear output head with an essentially frozen backbone, which often causes the output head to overfit while leaving the backbone insufficiently adapted for meaningful refinement. To address this, we introduce two modifications: (1) enforcing imbalanced optimization by setting the head's learning rate $\eta_{\text{head}}$ significantly lower than that of the backbone $\eta_{\text{bb}}$; and (2) adopting a staged strategy that first trains the head for $E_{\text{head}}$ epochs with the backbone frozen, followed by backbone fine-tuning for $E_0$ epochs with the head fixed ($E_{\text{head}} \ll E_0$). This asymmetric training scheme compels the backbone to absorb most gradient signals, progressively enhancing representation quality with limited data and achieving performance close to the upper bound. It is worth noting that our strategy is opposite to those of LAE (Gao et al., 2023) and SLCA (Zhang et al., 2023), where backbone

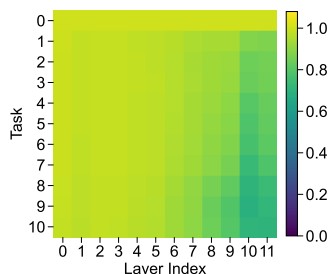

Figure 5: CKA (Kornblith et al., 2019) visualization shows representation changes of the first session classes across layers in CL.

updates are suppressed to mitigate forgetting. This contrast highlights the unique characteristics of audio backbones, where selectively encouraging backbone adaptation is critical for effective transfer without incurring catastrophic forgetting.

**Later Layer LoRA.** The empirical analysis in Fig. 3(c) suggests that the first-session performance may improve when adaptation is restricted to deeper layers, i.e., freezing early layers while tuning later ones. This aligns with the hierarchical structure of audio models: shallow layers tend to encode domain-general time–frequency and acoustic patterns (Niizumi et al., 2021; Cramer et al., 2019), whereas deeper layers capture higher-level semantic abstractions that are more task-specific (Liu et al., 2020a; Baevski et al., 2020). Further supported by the centered kernel alignment (CKA) visualization in Fig. 5, which shows that the model tends to stay relatively stable in the early layers, we freeze encoder layers 1 through $L_{\text{tune}} - 1$ and only adapt layers $l \geq L_{\text{tune}}$. The boundary layer $L_{\text{tune}}$ is determined via representation shift analysis during full fine-tuning: we select the shallowest layer whose CKA deviation from the pretrained model exceeds a threshold $\rho_{\text{layer}}$. Implementation details are provided in Algorithm 1 and elaborated in Sec. B.

Combining the above strategies, we apply LoRA to the tunable layers $l \in [L_{\text{tune}}, L]$, where $L$ denotes the total layer number. For each such layer, the adapted weight is given by:

$$W_1^l = W_0^l + A_1^l B_1^l, \quad \text{for } L_{\text{tune}} \leq l \leq L, \tag{1}$$

where $W_0^l \in \mathbb{R}^{d \times d}$ is the pretrained weight, and $A_1^l \in \mathbb{R}^{d \times r}$, $B_1^l \in \mathbb{R}^{r \times d}$ are trainable low-rank matrices with rank $r \ll d$. This design allows us to efficiently refine task-relevant semantic features while preserving general audio representations, yielding robust FSA without full model tuning.

**Analytic Classifier.** To maximally leverage the stabilization of prior representations and prevent accumulated biases from a trainable head $h^t(\cdot)$, we adopt an exemplar-free recursive analytic classifier (McDonnell et al., 2023) for final predictions. Given a random projector $W_{\text{proj}} \in \mathbb{R}^{D \times D_{\text{proj}}}$ to enhance feature discriminability (Tran et al., 2025), we can obtain the projected feature matrix $\hat{Z}_t = W_{\text{proj}} Z_t = [\hat{z}_{i,t}, \cdots, \hat{z}_{N_t,t}]^\top \in \mathbb{R}^{N_t \times D_{\text{proj}}}$ and corresponding one-hot label matrix $Y_t \in \mathbb{R}^{N_t \times |\mathcal{Y}_t|}$, the autocorrelation matrix $R^t$ is initialized with a regularization term $\gamma > 0$, i.e., $R_t = (\hat{Z}_t^\top \hat{Z}_t + \gamma I)^{-1}$. This is then recursively updated using the Woodbury identity:

$$R_t = R_{t-1} - R_{t-1} \hat{Z}_t^\top (I + \hat{Z}_t R_{t-1} \hat{Z}_t^\top)^{-1} \hat{Z}_t R_{t-1}. \tag{2}$$

Classifier weights $\hat{W}_t$ are then updated via a closed-form rule:

$$\hat{W}_t = \hat{W}_{t-1} - R_t \hat{Z}_t^\top \hat{Z}_t \hat{W}_{t-1} + R_t \hat{Z}_t^\top Y_t. \tag{3}$$

In inference, a new feature $z_{i,t}$ is classified by: $\hat{y}_{i,t} = \phi_t(W_{\text{proj}} z_{i,t}) = \hat{z}_{i,t} \hat{W}_t$. This design allows continual, exemplar-free updates to decision boundaries while preserving alignment with the stabilized representation space, ensuring robust and non-destructive learning.

## 3.3 Adaptive Multi-Session Subspace-Orthogonal PEFT

FSA (Zhuang et al., 2022; McDonnell et al., 2023) mitigates catastrophic representation shifts in the audio domain by refining pretrained representations during the first task. While this is often sufficient for coarse-grained datasets with minimal domain gap, it severely constrains backbone plasticity, limiting its applicability in fine-grained scenarios. In such settings, bridging the semantic mismatch between pretraining and downstream tasks requires learning new representations beyond those established in the first session. To address this challenge, we propose *Adaptive Multi-Session Subspace-Orthogonal PEFT* that enables adequate adaptation across multiple sessions while avoiding destructive interference with previously acquired representations. Our key idea is to leverage data from multiple tasks to reshape the representation space, aligning it with downstream semantics under large domain gaps, while simultaneously preserving the geometry of the previous representation space to maintain compatibility with the analytic classifier. Once further backbone adaptation offers diminishing returns, we freeze the backbone for long-term stability, entering Stage 3 ($t > T_3$).

**Multi-Session Adaptation (MSA).** We extend FSA to the multi-session setting by introducing session-specific LoRA (Hu et al., 2021), allowing each session to adapt the backbone while retaining the parameters from previous ones. Specifically, for each session $t \in (1, T_3]$, we augment the base weights $W_0$ with new low-rank updates $A_t B_t$, forming the session-specific parameters

$W_t = W_0 + \sum_{\tau=0}^{t-1} B_\tau A_\tau + B_t A_t$, where $\{A_\tau, B_\tau\}_{\tau=0}^{t-1}$ are frozen to prevent retroactive interference. However, updates may still misalign with earlier representations, leading to catastrophic forgetting on decision boundaries. To counter this, we aim to ensure that the backbone update $g_{\text{update}}$ does not significantly alter the representations of past tasks. Formally, for any old sample $x_{i,\tau} \in \mathcal{X}_\tau$ with $\tau < t$, the representation change should satisfy

$$\Delta f_t(x_{i,\tau}) = -\eta_{\text{bb}} \, g_{\text{update}}^\top x_{i,\tau} \approx 0, \tag{4}$$

where $\eta_{\text{bb}}$ is the backbone learning rate. To achieve this, we constrain $g_{\text{update}}$ to lie in the null space of the subspace spanned by previous representations. Specifically, let $g_{\text{original}}$ denote the gradient computed from the cross-entropy loss $\mathcal{L}_{\text{ce}}$ between the current model predictions $g_t(f_t(\mathcal{X}_t))$ and their labels $\mathcal{Y}_t$. We then define the projected update as

$$g_{\text{update}} = P_{\mathcal{U}_t} g_{\text{original}} = P_{\mathcal{U}_t} \nabla_\theta \mathcal{L}_{\text{ce}}(g_t(f_t(\mathcal{X}_t)), \mathcal{Y}_t), \tag{5}$$

where $P_{\mathcal{U}_t} = U_t U_t^\top$, and $U_t$ is an orthonormal basis spanning the null subspace $\mathcal{U}_t$ of all previously acquired representations. This projection ensures that adaptation updates minimally affect old samples, thereby preventing distortion of learned representations and maintaining stability in CL.

To compute the null space $\mathcal{U}_t$, a naive way would require storing all historical features from $\mathcal{X}_1, \ldots, \mathcal{X}_{t-1}$ (Wang et al., 2021; 2024b), resulting in extensive storage overheads. Inspired by LoRA Subtraction (Liu & Chang, 2025; Ilharco et al., 2023), we instead construct an *unlearned model* by subtracting all previous LoRA parameters $W_t^{\text{unlearn}} = W_0 - \sum_{\tau=0}^{t-1} A_\tau B_\tau$.

For computational efficiency, we then compute the *uncentered* covariance matrix of the current session's features by $X_t^{\text{ucov}} = f_t^{\text{unlearn}}(\mathcal{X}_t)^\top f_t^{\text{unlearn}}(\mathcal{X}_t) \in \mathbb{R}^{D \times D}$, which shares the same principal subspace with $f_t^{\text{unlearn}}(\mathcal{X}_t)^\top \in \mathbb{R}^{N_t \times D}$ where $f_t^{\text{unlearn}}(\cdot)$ denotes the frozen feature extractor using $W_t^{\text{unlearn}}$, and $D$ is the feature dimension. Through performing singular value decomposition (SVD) on $X_t^{\text{ucov}}$, we obtain its eigendecomposition: $X_t^{\text{ucov}} = U_t^l \Lambda_t^l (U_t^l)^\top$, and define the layer-wise projection operator as $P_{\mathcal{U}_t^l} = U_t^{l,(1:m)} (U_t^{l,(1:m)})^\top$, where

$$m = \arg\max_m \frac{\sum_{i=1}^m \lambda_i}{\sum_{i=1}^D \lambda_i} > \rho_{\text{svd}}, \quad \lambda_i : i\text{-th max diagonal entry of } \Lambda_t^l. \tag{6}$$

Despite orthogonal projection, continual adaptation can still accumulate shifts that subtly degrade early-session compatibility, especially when class sizes are small. We empirically find that controlling the cumulative number of seen samples offers a good stability–plasticity trade-off. Specifically, we define the threshold as $T_3 = \arg\max_{T_3} \sum_{i=0}^{T_3} N_t > N_{\text{stop}}$, where $N_{\text{stop}}$ is a threshold controlling the stability–plasticity trade-off. Once the backbone stabilizes, we transition to Stage 3 by freezing the backbone to preserve learned knowledge and update only the analytic classifier thereafter.

**Boundary-Aware Regularization.** While MSA preserves representations of previous sessions, it may reduce the plasticity needed for learning new sessions (Liu & Chang, 2025), especially when new- and old-class representations become entangled. This entanglement can cause updated decision boundaries to confuse past representations. To enhance separation, we introduce a boundary-aware regularization that encourages intra-class compactness and inter-class margin enlargement.

Specifically, for each input $x_{i,t}$ at session $t$, we generate perturbed samples $\tilde{x}_{i,t}^k = \mathcal{Q}(x_{i,t}, r_T, r_F)$ using time-frequency masking (Park et al., 2019), where $\mathcal{Q}$ randomly masks time and frequency dimensions with ratios $r_T \leq R_T$ and $r_F \leq R_F$, producing $N_p$ perturbations per input. To detect boundary-prone samples, we examine whether these perturbations are consistently misclassified by a temporary model $\theta_{\text{temp}}$, which frozen the backbone from the previous model $\theta_{t-1}$ with only the classification head adjusted to session $t$. We include such samples in the boundary set $\mathcal{B}_t$:

$$\mathcal{B}_t = \Big\{ \bar{f}_{t-1}(x_{i,t}) \mid \frac{1}{N_P} \sum_{k=1}^{N_P} \mathbf{1}\big[\hat{y}_{\theta_{\text{temp}}}(\tilde{x}_{i,t}^k) \neq y_{i,t}\big] > \rho_p \Big\}. \tag{7}$$

where $\mathbf{1}(\cdot)$ is the indicator function, and $\rho_p$ is the misclassification threshold.

To regularize representations, we define the following loss for each clean input and its perturbations:

$$\mathcal{L}_{\text{reg}}(i) = \max\left(0, \quad \delta + \frac{1}{|\mathcal{S}_i|} \sum_{u \in \mathcal{S}_i} \big\|\bar{f}_t(u) - \bar{\mu}(x_c)\big\|_2^2 - \min_{b \in \mathcal{B}_t} \big\|\bar{f}_t(x_{i,t}) - b\big\|_2^2 \right), \tag{8}$$

where $\mathcal{S}_i = \{x_{i,t}, \tilde{x}_{i,t}^1, \ldots, \tilde{x}_{i,t}^{N_P}\}$ and $\bar{\mu}(x_c)$ is the centroid of class $c$. This pulls features toward their class center and pushes them away from nearby boundary points, mitigating future confusion.

# 4 EXPERIMENT

## 4.1 EXPERIMENTAL SETTING

We adopt EAT (Chen et al., 2024), a general-purpose self-supervised audio backbone pretrained on AudioSet-2M ($\sim$ 5000 hours) (Gemmeke et al., 2017), as the default implementation. EAT is a spectrogram-based masked prediction model following a ViT architecture with 12 Transformer (Vaswani et al., 2017) blocks. We benchmark performance across 3 coarse-grained and 3 fine-grained datasets, spanning environmental sounds, speech, and music. All datasets are randomly split into an 8:2 ratio for training and testing. Additional details are provided in Sec. D.

**Coarse-grained Audio CL Benchmark. ESC-50** (Piczak, 2015) consists of 2000 samples from 50 classes (5 seconds each), covering numerous environmental sounds such as barking, rain, doorbell, and sawing. **UrbanSound8K (US8K)** (Salamon et al., 2014) contains 8732 samples from 10 urban sound classes (up to 4 seconds each), representing typical city sounds such as air conditioner and car horn. **Speech Command v2 (SC2)** (Warden, 2018) includes 105k one-second recordings from 35 speech classes for keyword recognition. For CL, ESC-50 is split into 10 sessions with 5 classes each, US8K into 5 sessions with 2 classes each, and SC2 into 7 sessions with 5 classes each. These datasets are either domain-matched with the pretrained model or well-adapted in the first session.

**Fine-grained Audio CL Benchmark. TIMIT** (Garofolo et al., 1993) is a classic speech corpus with 630 speakers from 8 U.S. dialects, each reading 10 phonetically rich sentences, totaling about 6300 utterances. We reformulate TIMIT as a continual speaker identification benchmark with 2 settings: **TIMIT-2** (315 tasks with 2 speakers each) and **TIMIT-3** (210 tasks with 3 speakers each). **VocalSet** (Wilkins et al., 2018) is a curated dataset for singing technique recognition and singer identification. It contains about 3560 clips ($\approx$10 hours) from 20 singers, covering 16 vocal techniques such as vibrato and belt. To balance the dataset, we randomly sample 79 clips per technique (64 for training and 15 for testing), and further split them into 8 sessions with 2 classes each.

**Baseline Methods.** We adopt the EAT (Chen et al., 2024) backbone, pretrained on AudioSet-2M (Gemmeke et al., 2017) with $\sim$5,000 hours of audio. EAT follows the ViT design with 12 Transformer (Vaswani et al., 2017) blocks. To validate the effect of pretrained audio CL models, we consider state-of-the-art (SoTA) baselines developed for vision domain, including (1) PEFT-based methods such as L2P (Wang et al., 2022c), DualPrompt (Wang et al., 2022b), S-Prompt++ (Wang et al., 2022a), LoRASub (Liu & Chang, 2025), and HiDe-PET (Wang et al., 2023; 2025), and (2) statistics-based methods, such as Nearest-Prototype Classification (NPC) (Rebuffi et al., 2017b), RanPAC (McDonnell et al., 2023) and ACL (Zhuang et al., 2022).

## 4.2 EXPERIMENTAL RESULTS

**Overall Performance.** We evaluate PACE against SoTA methods across six audio CL benchmarks, using the self-supervised pretrained audio model EAT (Chen et al., 2024). In Table 2, PACE consistently outperforms all baselines, achieving the best performance across both coarse and fine benchmarks. We also conduct additional benchmarks on non-human audio and cross-domain evaluation, and further validate the effectiveness of PACE with another self-supervised pretrained audio model SSLAM (Alex et al., 2025), as shown in Sec. E.3 and Sec. E.5, respectively.

On **coarse-grained benchmarks** (ESC-50, US8K, SC2), joint training with LoRA yields strong results by fully leveraging task-specific supervision, highlighting the potential backbone plasticity under non-continual conditions. However, *prompt-based methods* such as L2P, DualPrompt, and S-Prompt++ perform poorly in CL due to their reliance on shared prompt keys, which are highly vulnerable to forgetting. Their hybrid use of task-shared and task-specific components often induces representation shifts, sometimes performing worse than naive PEFT. HiDe-PET partially addresses classifier forgetting via feature replay, but its effectiveness is limited as the stored features themselves suffer continual representation shifts. LoRASub mitigates drift to some extent but still inherits continual classifier degradation and requires parameter expansion over long task sequences (e.g., TIMIT). In contrast, *statistics-based methods* such as RanPAC and ACL freeze the backbone and rely on FSA with analytic classifier, offering more robust performance by avoiding drift. Nonetheless, they encounter a ceiling due to representation saturation, particularly in early adaptation. PACE overcomes this limitation via layer-aware tuning and adaptive subspace-orthogonal regularization,

Table 2: A sample of average top-1 accuracy (%) of different methods on six audio CL benchmarks.

| Method | Coarse-Grained | | | Fine-Grained | | |
|---|---|---|---|---|---|---|
| | ESC-50 | US8K | SC2 | TIMIT-2 | TIMIT-3 | VocalSet |
| *Naive Methods* | | | | | | |
| EAT (LoRA) + Joint Training | 96.50 | 98.07 | 95.91 | 95.22 | 95.22 | 76.65 |
| EAT (Frozen) + Linear probe | 40.50 | 49.99 | 32.84 | 2.30 | 4.92 | 13.82 |
| EAT (Prompt) + Linear Probe (Lester et al., 2021) | 58.75 | 44.98 | 41.54 | 1.67 | 1.43 | 22.04 |
| EAT (Adapter) + Linear Probe (Houlsby et al., 2019) | 59.25 | 38.76 | 55.33 | 6.22 | 14.12 | 16.45 |
| EAT (LoRA) + Linear Probe (Hu et al., 2021) | 64.00 | 49.68 | 30.56 | 0.00 | 2.62 | 15.79 |
| *PEFT-Based CL Methods* | | | | | | |
| L2P (Wang et al., 2022c) | 39.50 | 38.75 | 14.70 | 1.50 | 2.53 | 20.39 |
| DualPrompt (Wang et al., 2022b) | 57.00 | 42.40 | 21.92 | 5.87 | 10.00 | 12.50 |
| S-Prompt++ (Wang et al., 2022a) | 55.00 | 42.57 | 27.23 | 6.43 | 8.25 | 17.76 |
| HiDe-Prompt (Wang et al., 2023) | 83.75 | 79.89 | 40.10 | 47.78 | 49.60 | 48.36 |
| HiDe-LoRA (Wang et al., 2025) | 88.75 | 76.48 | 33.66 | 47.30 | 49.60 | 46.05 |
| HiDe-Adapter (Wang et al., 2025) | 82.75 | 78.03 | 33.71 | 7.14 | 12.22 | 49.67 |
| LoRASub (Liu & Chang, 2025) | 57.50 | 57.81 | 34.24 | 0.00 | 0.00 | 24.01 |
| *Statistics-Based CL Methods* | | | | | | |
| Nearest Class Mean (NCM) (Rebuffi et al., 2017b) | 33.25 | 36.09 | 9.30 | 6.90 | 6.83 | 32.89 |
| Nearest Class Mean (NCM) w/ FSA | 49.00 | 42.44 | 57.60 | 23.97 | 34.68 | 34.53 |
| ACL (High-Order) (Zhuang et al., 2022) | 90.00 | 95.98 | 80.29 | 75.56 | 75.56 | 62.50 |
| RanPAC (High-Order) w/o FSA | 92.50 | 96.49 | 81.22 | 75.87 | 75.87 | 61.51 |
| RanPAC (High-Order) (McDonnell et al., 2023) | 92.25 | 97.08 | 90.53 | 85.63 | 89.92 | 62.82 |
| PACE (Ours) | **95.75** | **97.49** | **91.87** | **90.95** | **94.05** | **69.08** |

Table 3: Ablation results of our improved FSA on coarse-grained datasets.

| Method | ESC-50 | US8K | SC2 |
|---|---|---|---|
| w/o FSA | 92.50 | 96.49 | 81.22 |
| Naive FSA | $92.25^{-\ 0.27\%}$ | $97.08^{+\ 0.61\%}$ | $90.53^{+11.46\%}$ |
| Low Learning Rate | $93.75^{+\ 1.35\%}$ | $97.35^{+\ 0.89\%}$ | $90.95^{+11.98\%}$ |
| Learning & Freeze | $94.50^{+\ 2.16\%}$ | $97.38^{+\ 0.92\%}$ | $91.30^{+12.41\%}$ |
| Our FSA | $95.75^{+\ 3.51\%}$ | $97.49^{+\ 1.04\%}$ | $91.87^{+13.11\%}$ |

Table 4: Ablation results of our key components on fine-grained datasets.

| Method | TIMIT-2 | TIMIT-3 | VocalSet |
|---|---|---|---|
| Ours | 90.95 | 94.05 | 69.08 |
| Ours w/o FSA | $75.87^{-16.57\%}$ | $75.87^{-19.35\%}$ | $61.51^{-10.97\%}$ |
| Ours w/o MSA | $85.63^{-\ 5.86\%}$ | $89.92^{-\ 4.40\%}$ | $62.82^{-\ 9.06\%}$ |
| Ours w/o $\mathcal{L}_{reg}$ | $89.21^{-\ 1.91\%}$ | $93.73^{-\ 0.34\%}$ | $66.78^{-\ 3.33\%}$ |
| Ours w/o GP | $88.01^{-\ 3.23\%}$ | $89.05^{-\ 5.31\%}$ | $58.55^{-15.26\%}$ |

enhancing representation plasticity while maintaining stability. As a result, it consistently narrows the gap to the joint training upper bound (e.g., 0.75% on ESC-50, 0.58% on US8K).

On **fine-grained benchmarks** (TIMIT-2&3, VocalSet), we observe significantly lower performance of all baselines, including joint training, highlighting the inherent domain gap in these tasks. TIMIT in particular suffers from instability due to the large number of classes (up to 630). *Prompt-based methods* consistently fail in these settings. *Statistics-based methods*, while more stable, leave a substantial gap relative to joint training. In contrast, PACE substantially narrows this gap, as FSA-based statistical methods alone are insufficient to effectively align representations with downstream domains. By augmenting FSA with MSA, while simultaneously mitigating representation forgetting, combined with audio-specific PEFT and tailored perturbation loss, PACE achieves more discriminative and stable representations. Consequently, it demonstrates stronger alignment between pretraining and downstream tasks in fine-grained scenarios, where the performance gap from joint training is much smaller (4.27% and 1.17% on TIMIT-2 and TIMIT-3, 7.57% on VocalSet).

**Ablation Studies.** Since coarse-grained tasks are well handled by FSA alone, but fine-grained tasks require further adaptation via MSA, we evaluate FSA design choices on coarse datasets (Table 3) and ablate all core components on fine-grained datasets (Table 4).

Table 3 demonstrates that our improved FSA (see Sec. 3.2) outperforms the naive FSA by combining two key design choices: restricted head learning (via a low learning rate and early freezing of the classification head) and controlled adaptation of later transformer layers. To further examine the sensitivity of the layer-freezing threshold, we select $\rho_{layer} = 0.94$ through grid search on fine-grained datasets. This threshold determines which layers remain frozen during adaptation (see Fig. 6(a)). As shown in Fig. 6(b), freezing exactly these layers yields the best performance, confirming the importance and effectiveness of later-layer adaptation. In addition to FSA, Table 4 shows that the full PACE model arises from the complementary contributions of its core components, including MSA

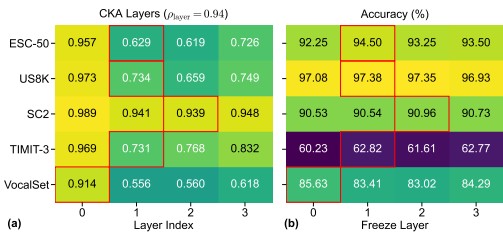

Figure 6: Sensitivity of the frozen-layer number ($L_{\text{tune}}$) within our FSA strategy.

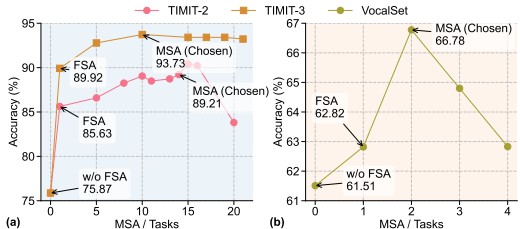

Figure 7: Sensitivity of the adaptation-session number ($L_3$) within our MSA strategy.

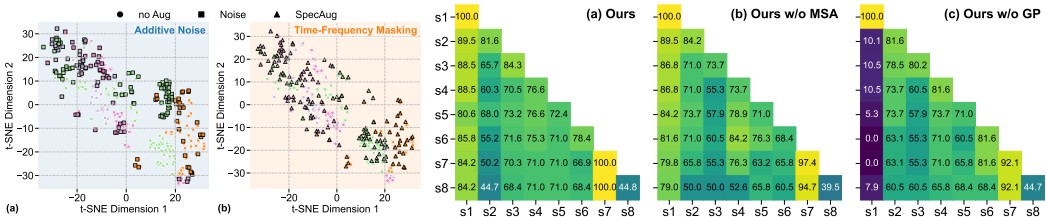

Figure 8: t-SNE visualization comparing different perturbation effects.

Figure 9: Heatmap visualization of the accuracy across sessions: (a) Ours, (b) Ours w/o MSA, and (c) Ours w/o GP.

(see Sec. 3.3), gradient projection, and boundary-aware regularization ($\mathcal{L}_{\text{reg}}$). Each component provides a distinct benefit, and removing any of them leads to a clear performance drop on fine-grained tasks. We additionally examine the sensitivity of the number of adaptation sessions in our MSA strategy. In Fig. 7, performance remains stable across a broad range of adaptation-session settings, demonstrating that PACE is robust to this hyperparameter and that our early-stopping mechanism effectively identifies an appropriate adaptation horizon, as further detailed in Sec. E.1. We also examine the remaining hyperparameter sensitivities in Sec. E.6.

**Visualizations.** To aid in the better understanding of our method, we provide additional visual analyses. In Fig. 8, we compare perturbation effects using t-SNE: additive noise (see Fig. 8(a)) significantly distorts the data manifold, whereas time–frequency masking (see Fig. 8(b)) preserves local neighborhood structure and class consistency, making it suitable for boundary-aware regularization. Fig. 9 visualizes accuracy across sessions, highlighting the stabilizing effect of MSA and GP. Without GP (see Fig. 9(c)), the model suffers severe catastrophic forgetting; for example, after completing Session 8, the accuracy on Session 1 classes drops from its initial 100% to 7.9%, indicating substantial representation drift and boundary collapse. In contrast, full PACE (see Fig. 9(a)) maintains high accuracy throughout, demonstrating that GP and boundary-aware regularization effectively constrain cross-session interference and preserve earlier knowledge.

## 5 CONCLUSION

While PTMs with PEFT have enabled substantial progress in vision CL, their direct application to audio faces fundamental challenges due to severe upstream–downstream mismatch. Through systematic benchmarking, we uncover unique obstacles in audio CL, such as representation saturation in early adaptation and representation shift in fine-grained scenarios. To address these issues, we propose PACE, a unified framework that combines selective first-session adaptation, adaptive subspace-orthogonal PEFT, and boundary-aware perturbations to enhance representation alignment and maintain an appropriate plasticity–stability trade-off. PACE achieves state-of-the-art performance across six diverse audio CL benchmarks. Although it requires slightly more adaptation time than RanPAC, the overall training cost remains substantially lower than prior PEFT-based baselines, demonstrating its effectiveness in both coarse- and fine-grained scenarios. Beyond technical contributions, our work offers a foundation for robust continual adaptation of pretrained audio models, with broad relevance to real-world applications in speech recognition, audio captioning, smart homes, environmental sound understanding, and so on.

**Acknowledgment.** This work was supported by the NSFC Project No. 62406160 and Beijing Natural Science Foundation L247011 and Beijing Major Science and Technology Project No. Z251100008425003.

**Ethics statement.** I acknowledge that I and all co-authors of this work have read and commit to adhering to the ICLR Code of Ethics.

**Reproducibility statement.** We have included the source code with clear instructions, and will release them upon acceptance.

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

## A STATEMENT

**Acknowledgment.** This work was supported by the NSFC Project No. 62406160 and Beijing Natural Science Foundation L247011.

**Ethics statement.** I acknowledge that I and all co-authors of this work have read and commit to adhering to the ICLR Code of Ethics.

**Reproducibility statement.** We have included the source code with clear instructions, and will release them upon acceptance.

**Large language models assistance.** Large language models were used to polish the manuscript. The authors have thoroughly reviewed and edited all content and take full responsibility for the published work.

## B PSEUDOCODE OF IMPROVED FIRST SESSION ADAPTATION

---

**Algorithm 1:** Improved First Session Adaptation

---

**Require:** Pretrained backbone $f_0$, $E_0$, $E_{\text{head}}$, CKA threshold $\rho_{\text{layer}}$, learning rates $\eta_{\text{bb}} > \eta_{\text{head}}$

1: **Stage A (detect layer boundary):**
2: $\theta_1^{\text{pre}} \leftarrow \theta_0$
3: **for** epoch $= 1, \cdots, E_0$ **do**
4:     $\theta_1^{\text{pre}} \leftarrow \theta_1^{\text{pre}} - \eta_{\text{bb}} \nabla_\theta \mathcal{L}_{\text{ce}}(h_{\text{ce}}(f_1^{\text{pre}}(\mathcal{X}_1)), \mathcal{Y}_1)$          {PEFT on all layers}
5: **end for**
6: **for** $k = 1, \cdots, L$ **do**
7:     $s^k \leftarrow \text{CKA}(f_1^{\text{pre}}(\mathcal{X}_1)^k, f_0(\mathcal{X}_1)^k)$
8: **end for**
9: $k^\star \leftarrow \arg\max_k [\, s^k < \rho_{\text{layer}} \,]$
10: **Stage B (head warm-up, $\eta_{\textbf{head}}$):**
11: **for** epoch $= 1, \cdots, E_{\text{head}}$ **do**
12:     $h_1 \leftarrow h_1 - \eta_{\text{head}} \nabla_h \mathcal{L}_{\text{ce}}(h_1(f_0(\mathcal{X}_1)), \mathcal{Y}_1)$
13: **end for**
14: **Stage C (backbone adaptation, $\eta_{\textbf{bb}}$):**
15: Freeze $h_{\text{ce}}, \theta_0$; $\theta_1 \leftarrow \theta_0 + \theta_1^{\text{LoRA}}$
16: **for** epoch $= 1, \cdots, E_0$ **do**
17:     $\theta_1 \leftarrow \theta_1 - \eta_{\text{bb}} \nabla_\theta \mathcal{L}_{\text{ce}}(h_1(f_1(\mathcal{X}_1)), \mathcal{Y}_1)$          {only layers $k^\star+1(L_{\text{tune}}){:}L$ in $f_{\theta_1}$ trainable}
18: **end for**
19: **Stage D (analytic phase):**
20: Freeze $\theta_1$; Discard $h_1$; Initialize statistics on $\phi(\cdot) = f_1(\cdot)$; Continue with Sec. 3.3.

---

Algorithm 1 outlines the improved first-session adaptation procedure, designed to establish effective yet stable representation transfer in FSA for audio CL. Stage A performs a lightweight PEFT update across all layers to probe sensitivity, after which CKA is used to measure the similarity between $f_1^{\text{pre}}$ and the pretrained backbone $f_0$ at each layer. The boundary layer $L_{\text{tune}} = k^\star + 1$ is identified as the point where representation similarity tend to drop below the threshold $\rho_{\text{layer}}$, indicating where proper adaptation should begin. Stage B then train a restricted classifier head $h_1$ with a smaller learning rate $\eta_{\text{head}}$ and training epoch $E_{\text{head}}$. Stage C adapts only the deeper layers beyond $k^\star$ with LoRA-style updates and a larger learning rate $\eta_{\text{bb}}$, while freezing shallow layers and the head to preserve pretrain knowledge and enhance backbone adaption. Finally, Stage D freezes the adapted backbone, discards the temporary head, and initializes statistics for the analytic phase (see Sec. 3.3).

## C RELATED WORKS

**Continual learning (CL)** enables models to learn continuously while mitigating catastrophic forgetting of previously learned knowledge (Wang et al., 2024a). Most research in CL has focused on the vision domain, with representative regularization-based (Kirkpatrick et al., 2017; Zenke et al., 2017), replay-based (Rolnick et al., 2019; Liu et al., 2020b), and architecture-based approaches (Douillard

et al., 2022; Kanakis et al., 2020). More recently, CL with pretrained models (PTMs) has attracted growing attention, as PTMs provide strong general-purpose initialization (Zhou et al., 2024a) and obviate the need for training from scratch. This paradigm typically leverages parameter-efficient fine-tuning (PEFT) (Rebuffi et al., 2017a; Li & Liang, 2021; Hu et al., 2021) strategies by introducing lightweight modules on top of frozen backbones (Wang et al., 2022c;b; Tran et al., 2025; Le et al., 2024; Gao et al., 2023; Zhao et al., 2024), or combines them with regularization principles (Liu & Chang, 2025; Liang & Li, 2024; Zhou et al., 2024b). In the vision domain, PTMs encode relatively stable and semantically well-structured representations (Janson et al.), leading to only mild representational shift across learning sessions (see Fig. 1(b)), which makes PEFT-based CL highly effective. However, the effectiveness of this paradigm in the audio domain remains underexplored and calls for systematic investigation.

**Audio recognition** covers tasks such as speech recognition (Abdel-Hamid et al., 2014; Bai et al., 2024), acoustic event detection (Yuan et al., 2025; Li et al., 2025b), and music understanding (Li et al., 2024b). Recent pretrained models (Gong et al., 2022; Chen et al., 2024; Alex et al., 2025; Liu et al., 2025; Chang et al., 2025) have achieved strong performance across these domains, and show potential for providing conditioning or regularization signals in audio generation tasks (Li et al., 2025a; 2024a; Jiang et al., 2025). Yet these advances usually assume offline training with full data access, overlooking the evolving, non-stationary distributions of real-world settings (Bhatt et al., 2024). Emerging studies on audio CL (Mulimani & Mesaros, 2025; Cappellazzo et al., 2024; Singh et al., 2024; Mo et al., 2023; Pian et al., 2023) often borrow from vision or use simplified settings, without tackling audio's unique challenges. Unlike images, audio depends on fine-grained spectral cues, making representations highly sensitive to distribution shifts and prone to catastrophic forgetting (see Fig. 1(a)). This calls for audio-specific CL strategies compatible with pretrained backbones and aligned with the spectral nature of audio.

## D  ADDITIONAL EXPERIMENTAL DETAILS

All experiments are conducted on an NVIDIA A800 GPU. The input size is $512 \times 128$. Each audio clip is truncated to the first 5.12 seconds, with a batch size of 24. We set $\eta_{bb} = 0.05$, $\eta_{head} = 0.01$ for all tasks, and the number of training epochs is selected via grid search. For each task, we report the average accuracy over all tasks, i.e., $\overline{\text{Acc}} = \frac{1}{T} \sum_{t=1}^{T} \text{Acc}_t$, as the primary measure of CL performance. For hypermarameters, we set $E_{head} = 1$, $\rho_{layer} = 0.94$ for improved First-Session Adaption, $\rho_{svd} = 0.99$, $N_{stop} = 220$ for Multi-Session Adaption, $N_p = 20$, $\rho_p = 0.3$, $\delta = 0.25$ for Boundary-Aware Perturbation and $D_{proj} = 8192$ for our Analytic Classifier. We also provide detailed statistics of the datasets along with their corresponding training epochs $E_0$ in Table 5.

Table 5: Statistics of datasets used in our experiments.

| Dataset | Epoch | Classes | Session | Total Train Samples | Total Test Samples |
|---|---|---|---|---|---|
| ESC-50 (Piczak, 2015) | 10 | 50 | 10 | 1600 | 400 |
| US8K (Salamon et al., 2014) | 15 | 10 | 5 | 8000 | 2000 |
| SC2 (Warden, 2018) | 1 | 35 | 7 | 84651 | 21178 |
| TIMIT-2 (Garofolo et al., 1993) | 30 | 630 | 315 | 5040 | 1260 |
| TIMIT-3 (Garofolo et al., 1993) | 30 | 630 | 210 | 5040 | 1260 |
| VocalSet (Wilkins et al., 2018) | 6 | 16 | 8 | 1216 | 304 |

## E  ADDITIONAL EXPERIMENTAL RESULTS

### E.1  STOP SESSION FOR MSA

As we set $N_{stop}$ to estimate the truncation point, we obtain a specific cutoff for each task, as shown in Table 6. To further assess the effectiveness of this task-level stopping mechanism, we conduct ablations on neighboring sessions around the chosen $T_3 = t^*$ (i.e., $t^* - 1$, $t^* - 2$, $t^* + 1$, $t^* + 2$). We observe that $t^*$ yields locally optimal results on most datasets (US8K, SC2, TIMIT-3, VocalSet),

| Dataset | Chosen $t^*$ | w/o FSA | FSA | $t^*-2$ | $t^*-1$ | $t^*$ | $t^*+1$ | $t^*+2$ |
|---|---|---|---|---|---|---|---|---|
| ESC-50 | 2 | 92.50 | 92.25 | N/A | **92.25** | 91.75 | 91.75 | 92.00 |
| US8K | 1 | 96.49 | 97.08 | N/A | N/A | **97.08** | 95.15 | 81.44 |
| SC2 | 1 | 81.22 | 90.53 | N/A | N/A | **90.53** | 89.46 | 81.14 |
| TIMIT-2 | 14 | 75.87 | 85.63 | 88.49 | 88.73 | 89.21 | **90.40** | 90.23 |
| TIMIT-3 | 10 | 75.87 | 89.92 | 92.78 | **93.73** | **93.73** | 93.41 | 93.41 |
| VocalSet | 2 | 61.51 | 62.82 | N/A | 61.51 | **66.78** | 64.80 | 62.83 |

Table 6: Results of different datasets under FSA and MSA (w/o $\mathcal{L}_{\text{reg}}$ w/o Improved FSA) with different adaptation sessions.

substantially outperforming both w/o FSA and vanilla FSA, while maintaining more stable performance across longer session sequences in realistic CL settings (TIMIT-2/3). Although $t^*$ is not always the global optimum on ESC-50 and TIMIT-2, applying MSA in a naive manner does not lead to significant degradation compared to the optimal, thus validating the necessity of our stopping strategy. Additionally, it is worth noting that we generally observe a decline in performance as the number of sessions for adaptation grows, which becomes more pronounced when the sessions are relatively small. We attribute this to the backbone having already sufficiently adapted to the downstream domain, such that the marginal gains from additional adaptation are outweighed by cumulative representation drifts that induce forgetting on earlier tasks, underscoring the importance of a stopping criterion.

### E.2 Learning vs. Forgetting

To clarify how stability and plasticity interact during continual updates, we evaluate multi-session adaptation (MSA) with and without gradient projection (GP) on the **VocalSet** benchmark. We report five key indicators: (1) *Forgetting* over sessions $t \leq T_3$, (2) *Plasticity* measured by the average maximum accuracy, (3) *Average Accuracy* before $T_3$, (4) *Average Accuracy* after $T_3$, and (5) *Backward Transfer (BWT)*. Results are shown in Table 7.

Table 7: Comparison of MSA with and without gradient projection on VocalSet.

| Setting | Forgetting | Plasticity | Ave Acc ($t \leq T_3$) | Ave Acc ($t > T_3$) | BWT |
|---|---|---|---|---|---|
| FSA | 27.63 | **92.10** | **64.48** | 60.52 | −10.90 |
| MSA w/o GP | 57.90 | **92.10** | 34.21 | 66.67 | −9.02 |
| MSA (w/ GP) | **23.69** | 88.16 | 64.47 | **70.62** | **−7.14** |

The results indicate that gradient projection greatly reduces forgetting while maintaining plasticity. Although both MSA variants reach comparable maximum accuracy, removing GP causes significant representation drift, resulting in high forgetting and degraded BWT. In contrast, GP-stabilized MSA maintains stable early-session performance and achieves higher accuracy in later sessions, demonstrating that projection effectively constrains destructive feature shifts while preserving adaptability.

### E.3 Additional Benchmarks on Non-Human Audio and Cross-Domain Evaluation

To more comprehensively assess CL performance under diverse audio conditions, two additional benchmarks are introduced: (1) a fine-grained non-human audio dataset and (2) a cross-domain sound–speech dataset. These benchmarks target scenarios characterized by substantial intra-class variation and distributional mismatch, which are central to the challenges addressed in this work.

**GTZAN: Fine-Grained Non-Human Audio.** Many instrument-related datasets (e.g., GTMUSIC (Sturm, 2012)) exhibit coarse semantic granularity. Preliminary analyses show that a non-music pretrained backbone, such as EAT, combined with a single-session adaptation step, can reach up to **99.8%** accuracy, indicating minimal distribution complexity and limited suitability for CL evalua-

tion. In contrast, the **GTZAN** dataset (Sturm, 2013) contains 10 musical genres with richer intra-class diversity, offering a more realistic fine-grained distribution shift that better reflects the evolving semantic structure in non-human audio CL.

**ESC–Speech: Cross-Domain Sound–Speech Benchmark.** To evaluate robustness under heterogeneous domain compositions, a synthetic cross-domain dataset named **ESC–Speech** is constructed by combining ESC-50 (environmental sounds) and SpeechCommands V2 (spoken words). This mixture introduces a substantial domain mismatch between non-verbal acoustic events and human speech, forming a challenging scenario for pretrained audio models.

**Experimental Protocol.** Both benchmarks are evaluated under CL settings tailored to their characteristics: GTZAN uses a 5-session split with 20 samples per class to capture its fine-grained musical variability, whereas ESC–Speech adopts a 10-session split with 50 samples per class to reflect its cross-domain (sound–speech) shifts. The results are presented in Table 8.

Table 8: CL results on more diverse benchmarks.

| Dataset | Attribute | L2P | HiDe-Prompt | RanPAC | PACE |
|---|---|---|---|---|---|
| GTZAN | Non-human Voice (Instrument) | 10.00 | 51.00 | 73.00 | **78.00** |
| ESC–Speech | Cross-domain (ESC + SC2) | 21.50 | 52.58 | 57.00 | **72.17** |

**Observations.** Across both benchmarks, PACE achieves the highest performance, demonstrating strong resilience to both intra-class variation (GTZAN) and cross-domain distribution shift (ESC–Speech). The musical-domain evaluation further reveals a substantial representational shift in pretrained audio models, which intensifies the difficulty of continual adaptation. Methods originally designed for vision-based CL (e.g., L2P) show limited effectiveness in such settings. In contrast, the multi-session adaptation strategy in PACE enables progressive alignment of the evolving semantic space, producing consistent improvements under both intra- and inter-domain distribution changes.

### E.4 COMPUTATIONAL COST AND TRAINING OVERHEAD

The computational overhead of PACE arises primarily from the early-stage backbone updates, which are required to obtain a semantically aligned representation. After the first few adaptation sessions, the method transitions to an analytic classifier together with lightweight subspace-orthogonal updates, resulting in highly efficient training for all subsequent sessions.

On coarse-grained benchmarks, improved FSA provides sufficient alignment, and no additional backbone updates are needed. In these settings, PACE achieves near joint-training performance with essentially the same computational cost as standard fine-tuning.

On fine-grained benchmarks, PACE introduces moderate overhead relative to RanPAC, but remains substantially more efficient than prompt-based approaches such as HiDe-Prompt, which repeatedly optimize prompts or low-rank adapters across all sessions. The results are reported in Table 9.

Table 9: Training time and training time ratios on fine-grained datasets. Training time is calculated on a single GPU and ratios are computed relative to RanPAC.

| Dataset | Training Time (sec/sample) | PACE / RanPAC | HiDe-Prompt / RanPAC |
|---|---|---|---|
| VocalSet | 0.22 | 1.22 | 5.44 |
| TIMIT-3 | 0.12 | 2.96 | 146.98 |
| TIMIT-2 | 0.31 | 3.13 | 124.19 |

These results show that PACE delivers significant gains in accuracy while introducing only modest computational overhead. In contrast, HiDe-Prompt incurs $5\times$–$40\times$ higher cost despite yielding inferior performance. Although our method requires more training time than RanPAC on fine-grained datasets, it still reaches an average speed of about 0.2 sec/sample on single GPU, which is highly efficient in practice. Overall, PACE provides a favorable efficiency–effectiveness trade-off, particularly in fine-grained continual learning scenarios.

### E.5 ADDITIONAL PRETRAINED BACKBONES

To evaluate the generality of PACE beyond a single pretrained checkpoint, we further benchmarked the framework using **SSLAM** (Alex et al., 2025), a recent source-aware pretrained audio model trained on polyphonic mixtures and sharing the same ViT backbone as EAT (Chen et al., 2024). Experiments were conducted on two coarse-grained and two fine-grained datasets, and PACE was compared against L2P, HiDe-Prompt, and RanPAC. Results are reported in Table 10.

Table 10: CL performance with the SSLAM backbone on coarse- and fine-grained datasets.

| Dataset | Granularity | Backbone | L2P | HiDe-Prompt | RanPAC | PACE |
|---------|-------------|----------|-----|-------------|--------|------|
| ESC-50 | Coarse | SSLAM | 40.50 | 82.00 | 95.75 | **96.25** |
| SC2 | Coarse | SSLAM | 15.24 | 37.85 | 88.59 | **90.39** |
| VocalSet | Fine | SSLAM | 17.76 | 47.22 | 63.83 | **68.42** |
| TIMIT-2 | Fine | SSLAM | 0.32 | 46.24 | 90.08 | **93.81** |

The results demonstrate that PACE consistently outperforms all baselines across both pretrained backbones and granularity levels. This indicates that the challenges identified in pretrained audio models, such as representation saturation and semantic misalignment, are not specific to EAT, but also arise in more recent models like SSLAM. Moreover, the strong performance across all settings highlights the robustness and general applicability of PACE as a CL framework for audio.

### E.6 HYPERPARAMETER SELECTION AND SENSITIVITY ANALYSIS

This work introduces several continuous variables in the design of PACE. To make these selection procedures explicit, we provide a comprehensive sensitivity analysis.

**SVD threshold $\rho_{svd}$.** Following prior PEFT-based CL studies (Liu & Chang, 2025; Wang et al., 2021), we directly adopt the same SVD energy threshold. Across all datasets, varying $\rho_{svd}$ within the range $[0.90, 0.99]$ yields nearly identical results, confirming that this hyperparameter mainly affects numerical rank selection and has minimal influence on the learning dynamics.

**Stopping threshold $N_{stop}$ and freezing threshold $\rho_{layer}$.** These two hyperparameters govern MSA and were jointly tuned on the fine-grained **VocalSet** and **TIMIT-3** datasets. Using only FSA, we searched values in $\{0.90, 0.92, 0.94, 0.96, 0.98, 1.00\}$ and selected the value that best balanced adaptation and stability. With $\rho_{layer}$ fixed, we tuned $N_{stop}$ using MSA without regularization. As indicated in Figs. 7(a) and 7(b), model performance plateaus between 200 and 250 gradient steps. Although $N_{stop} = 250$ slightly exceeds the optimal $T_3$ on VocalSet, it still improves over FSA by approximately 2%, suggesting that the method is resilient to small deviations.

**Validation of $\rho_{layer}$.** As shown in Fig. 6, the selected value $\rho_{layer} = 0.94$ successfully avoids premature freezing of adaptable deeper layers while preserving stable low-level feature extraction, especially for fine-grained datasets such as TIMIT-3.

**Sensitivity analysis.** We additionally examine the sensitivity of all remaining continuous hyperparameters. To improve readability, we present only the average performance and highlight the selected configurations. The results are reported in Tables 11 to 14.

Overall, PACE exhibits strong robustness across a wide range of hyperparameter configurations. The selected values serve as stable defaults and require minimal tuning, underscoring the practicality of the framework for CL across diverse audio scenarios.

### E.7 RATIONALE FOR TIME-FREQUENCY MASKING IN BOUNDARY-AWARE REGULARIZATION

The choice of time-frequency masking is motivated by the spectral structure of audio signals and the requirements of boundary-aware regularization. SpecAugment-style masking (Park et al., 2019) is adopted as a label-preserving perturbation applied directly in the time-frequency domain. By selectively removing narrow temporal or spectral bands, this operation introduces localized distortions to the spectrogram while preserving the global semantic identity of the signal. Such perturbations

Table 11: Sensitivity of the MSA stopping threshold $N_{stop}$ across three fine-grained datasets.

| Method | TIMIT-2 | TIMIT-3 | VocalSet | Avg. |
|---|---|---|---|---|
| **FSA** | 85.63 | 89.92 | 62.82 | **79.46** |
| **MSA w/ 150** | 88.25 | 92.78 | 62.82 | 81.28 |
| **MSA w/ 175** | 89.05 | 93.24 | 66.78 | 83.02 |
| **MSA w/ 200** | 88.49 | 93.33 | 66.78 | 82.87 |
| **MSA w/ 210** | 89.21 | 93.33 | 66.78 | 83.11 |
| **MSA w/ 220** | 89.21 | 93.73 | 66.78 | **83.24** |
| **MSA w/ 230** | 90.39 | 93.73 | 66.78 | 83.63 |
| **MSA w/ 240** | 90.23 | 93.73 | 64.80 | 82.92 |
| **MSA w/ 250** | 90.23 | 93.41 | 64.80 | 82.81 |

Table 12: Sensitivity of the layer-freezing threshold $\rho_{layer}$ across datasets..

| $\rho_{layer}$ | TIMIT-3 | SC2 | VocalSet | ESC-50 | US8K | Avg. |
|---|---|---|---|---|---|---|
| 0.90 | 83.41 | 90.96 | 62.82 | 94.50 | 97.38 | 85.01 |
| 0.91 | 83.41 | 90.96 | 62.82 | 94.50 | 97.38 | 85.01 |
| 0.92 | 85.63 | 90.96 | 62.82 | 94.50 | 97.38 | 86.26 |
| 0.93 | 85.63 | 90.96 | 62.82 | 94.50 | 97.38 | 86.26 |
| **0.94** | 85.63 | 90.96 | 62.82 | 94.50 | 97.38 | **86.26** |
| 0.95 | 85.63 | 90.54 | 62.82 | 94.50 | 97.38 | 86.17 |
| 0.96 | 85.63 | 90.54 | 62.82 | 92.25 | 97.38 | 85.72 |
| 0.97 | 85.63 | 90.54 | 60.23 | 92.25 | 97.38 | 85.21 |
| 0.98 | 85.63 | 90.54 | 60.23 | 92.25 | 97.08 | 85.15 |
| 0.99 | 85.63 | 90.53 | 60.23 | 92.25 | 97.08 | 85.14 |
| 1.00 | 85.63 | 90.53 | 60.23 | 92.25 | 97.08 | 85.14 |
| w/o | 85.63 | 90.53 | 60.23 | 92.25 | 97.08 | 85.14 |

Table 13: Sensitivity of the boundary ratio $\rho_{ratio}$ on the VocalSet benchmark.

| Ratio | 0.20 | 0.25 | 0.30 | **0.35** | 0.40 | 0.45 | 0.50 |
|---|---|---|---|---|---|---|---|
| Acc | 65.79 | 67.43 | 67.43 | **69.08** | 68.75 | 64.47 | 64.47 |

Table 14: Sensitivity of the augmentation strength on the VocalSet benchmark.

| Strength | 0.10 | 0.20 | 0.30 | 0.40 | **0.50** | 0.60 | 0.70 | 0.80 | 0.90 |
|---|---|---|---|---|---|---|---|---|---|
| Acc | 65.79 | 66.12 | 64.80 | 63.49 | **69.08** | 68.10 | 65.79 | 64.47 | 61.51 |

enable controlled exploration of the neighborhood around each instance, supporting the construction of boundary-prone samples without violating class consistency.

To assess whether the proposed regularization term $\mathcal{L}_{reg}$ affects robustness, we further evaluated models on **VocalSet** with and without time-frequency masking applied at test time. The results are summarized in Table 15. Models trained with $\mathcal{L}_{reg}$ exhibit substantially smaller performance degradation under masked test inputs (1.83% vs. 6.26%), indicating that boundary-aware regularization enhances robustness rather than compromising it.

### E.8 CASE STUDY FOR COARSE-GRAINED AND FINE-GRAINED AUDIO CL

To further clarify the differences between fine- and coarse-grained datasets, we provide case studies in Fig. 10. Using PEFT-FT on both types of datasets, we track the prediction dynamics of three categories from the first session across the following three sessions, while also reporting the maximum inter-class distance in the pretrained model (measured in the first session) and the average feature

Table 15: Evaluation of robustness to time-frequency masking (TFM) on the VocalSet dataset.

| Evaluation | Test w/o TFM | Test w/ TFM |
|---|---|---|
| Train w/o TFM | 66.78 | 60.52 |
| Train w/ TFM | 69.08 | 67.25 |

shift between sessions. It is evident from Fig. 10(a) that coarse-grained datasets exhibit relatively large spectral patterns (e.g., the harmonic structure in *Cat* or the periodic energy bursts in *Door wood knock*). Such pronounced differences yield large inter-class separations, making it easier to capture them within the first session, which is reflected in the near joint-training performance of our improved FSA in Table 2. Consequently, when applying naive PEFT-FT, the model can adapt without substantial semantic changes in the representations, which corresponds to smaller cross-session feature shifts compared to the inter-class distances. Notably, categories with highly distinctive structures (e.g., *Door wood knock*) are more resistant to forgetting under such naive adaptation.

When it comes to fine-grained scenarios, as shown in Fig. 10(b), we observe that different categories share highly similar spectral patterns. In this setting, the model is required to discriminate among 630 speakers, where time–frequency details alone are insufficient to yield discriminative information. Instead, the model must adjust its representations within each session, using only limited data, to shape more discriminative deep features. This results in relatively small inter-class separations (nearly half those of coarse-grained datasets) but necessitates larger representational changes. The ultimate impact is catastrophic: although the model initially maintains high plasticity and achieves good accuracy within the current session, naive PEFT-FT lacks the memory capacity to retain prior knowledge, causing severe misclassification in subsequent sessions. This highlights the necessity of our PACE method, which effectively leverages data across multiple sessions to continually adapt the network while simultaneously constraining prior representations to remain stable.

**(a) Coarse-grained Dataset (ESC-50)**
[Maximum class distance: 16.01, Shifting across sessions: 4.70]

Cow

☑☑✗✗  ☑✗✗✗  ☑☑✗✗

Cat

☑☑☑☑  ☑☑✗✗  ☑☑☑✗

Door wood knock

☑☑☑✗  ☑☑☑☑  ☑☑☑☑

**(b) Fine-grained Dataset (TIMIT)**
[Maximum class distance: 9.88, Shifting across sessions: 8.06]

Speaker: FAJW0

☑✗✗✗  ☑✗✗✗  ☑✗✗✗

Speaker: FALK0

☑✗✗✗  ☑✗✗✗  ☑✗✗✗

Speaker: FBJL0

☑✗✗✗  ☑✗✗✗  ☑✗✗✗

Figure 10: Case studies on a coarse-grained dataset (ESC-50) and a fine-grained dataset (TIMIT-3): linear amplitude spectrograms of three classes from the first session, along with their performance trajectories under naive PEFT-FT.

