# OpenReview forum: "PACE: Pretrained Audio Continual Learning"
_ICLR.cc/2026/Conference — ICLR 2026 Poster_

### Official Review · Reviewer_4KNg · 2025-10-18

**Soundness:** 3
**Presentation:** 2
**Contribution:** 3
**Rating:** 6
**Confidence:** 3

**Summary:**

This paper investigates the challenges of continual learning with pretrained audio models. Through a systematic benchmark, the authors identify that a mismatch between pretraining objectives and downstream tasks leads to representation saturation in coarse-grained scenarios and severe representation shifts in fine-grained ones. To address this, they propose PACE, a framework that improves upon First-Session Adaptation (FSA) with three main contributions: (1) a restricted adaptation strategy using later-layer LoRA to prevent saturation , (2) an adaptive multi-session, subspace-orthogonal PEFT to enable stable learning across tasks , and (3) a boundary-aware perturbation regularizer to enhance class separability. Experiments across six diverse audio benchmarks demonstrate that PACE substantially outperforms existing state-of-the-art methods.

**Strengths:**

- Problem Formulation and Analysis: The paper excels in its methodical diagnosis of the problem. The identification of distinct failure modes for coarse-grained (representation saturation) and fine-grained (representation shift) tasks is a key insight that strongly motivates the design of the PACE framework.
- Novelty of the Proposed Method: PACE is a novel combination of several well-reasoned ideas. The adaptive multi-session subspace-orthogonal PEFT is a clever solution to balance plasticity and stability, and the later-layer LoRA adaptation is well-justified by both empirical results and intuition about hierarchical representations in neural networks.

**Weaknesses:**

- Clarity of Experimental Reporting: The paper's primary weakness lies in the fragmented and unclear presentation of its ablation studies, which makes it difficult to systematically assess the contribution of each component of the proposed PACE method. The validation results are scattered between the main text and the appendix, and even within the main paper, the connection between experiments and claims is not always explicit. For example, the caption for Table 3 is too general, forcing the reader to cross-reference the methodology to understand that it validates the "restricted head learning" strategy. A consolidated "Ablation Studies" section that clearly and methodically evaluates each of PACE's innovations would significantly strengthen the paper's clarity and persuasive impact.
- Hyperparameter Sensitivity: The proposed method introduces several new hyperparameters (e.g., the CKA threshold, the SVD threshold, the MSA stopping criterion, boundary misclassification threshold, ...). While the values are provided, the paper lacks a sensitivity analysis. It would be beneficial to understand how robust the method's performance is to variations in these hyperparameters.

**Questions:**

Could the authors clarify the design choices and associated trade-offs of the boundary-aware perturbation mechanism? The method's reliance on the previous model to identify boundary samples seems to imply a memory overhead from storing past models, which could be a significant drawback in long-sequence CL scenarios. Furthermore, time-frequency masking is typically employed as data augmentation to encourage invariance by pulling perturbed samples toward their class centroid. By contrast, the proposed method uses these perturbations to define a boundary to push features away from. Could the authors elaborate on the rationale for this specific design and discuss whether it potentially compromises the model's learned robustness to these common audio transformations?

---

> ### Author Response · Authors · 2025-11-20
> **Response to Reviewer 4KNg (Part 1)**
>
> Thank you for your thoughtful and constructive feedback. Below we address your comment with a point-to-point response.
>
> ---
>
> ### W1: Clarifying and Consolidating Ablation Studies
>
> Thank you for pointing this out. To improve clarity, we have substantially reorganized the ablation studies in the revised manuscript (see $\color{blue} \text{\bf Section 4.2}$ and $\color{blue} \text{\bf Appendix Section E.9}$).
>
> ---
>
> ### W2: Hyperparameter Sensitivity
>
> We have added the following hyperparameter sensitivity to $\color{blue} \text{\bf Appendix Section E.6}$
>
> **SVD Threshold ($\rho_{\text{svd}}$)**.
> We adopt $\rho_{\text{svd}}$ directly from prior CL works [2, 3]. In our preliminary experiments, varying $\rho_{\text{svd}}$ across a broad range (0.90–0.99) yielded nearly identical performance across all datasets. As $\rho_{\text{svd}}$ only determines the numerical rank used for projection (not the learning dynamics), its sensitivity is minimal.
>
> **Stopping Threshold ($N_{\text{stop}}$) and Layer Freezing Threshold ($\rho_{\text{layer}}$)**.
> These two hyperparameters were selected through a joint grid search on fine-grained benchmarks **VocalSet** and **TIMIT-3** where MSA is essential.
>
> - **Select $\rho_{\text{layer}}$**:
>    Using only FSA (no MSA or regularization), we searched over values in ${0.90, 0.92, 0.94, 0.96, 0.98, 1.00}$ and selected the value that best balanced adaptation and stability.
>
> - **Select $N_{\text{stop}}$**:
>   With $\rho_{\text{layer}}$ fixed, we tuned $N_{\text{stop}}$ using MSA without regularization ($L_{\text{reg}}$). As shown below, performance plateaus in the range of 200–250 updates, with $N_{\text{stop}} = 250$ giving strong, robust results. Although this slightly overshoots the transition session $T_3$  for VocalSet (ideal stop at $T_3 = 2$), it still improves over FSA by around 2%, indicating the method's resilience.
>
> **Validation of $\rho_{\text{layer}}$**
> As shown below and $\color{blue} \text{\bf Fig. 6}$, $\rho_{\text{layer}} = 0.94$ avoids premature freezing of adaptable layers (especially in fine-grained tasks like TIMIT-3), while preserving stable low-level features.

---

> ### Author Response · Authors · 2025-11-20
> **Response to Reviewer 4KNg (Part 2)**
>
> We provide a detailed sensitivity analysis for these key hyperparameters.
>
> #### (1) Sensitivity of **MSA Stopping Threshold** $N_{\text{stop}}$
>
> | **Method**     | **TIMIT-2** | **TIMIT-3** | **VocalSet** | **Avg.**  |
> | -------------- | ----------- | ----------- | ------------ | --------- |
> | **FSA**        | 85.63       | 89.92       | 62.82        | **79.46** |
> | **MSA w/ 150** | 88.25       | 92.78       | 62.82        | 81.28     |
> | **MSA w/ 175** | 89.05       | 93.24       | 66.78        | 83.02     |
> | **MSA w/ 200** | 88.49       | 93.33       | 66.78        | 82.87     |
> | **MSA w/ 210** | 89.21       | 93.33       | 66.78        | 83.11     |
> | **MSA w/ 220** | 89.21       | 93.73       | 66.78        | **83.24** |
> | **MSA w/ 230** | 90.39       | 93.73       | 66.78        | 83.63     |
> | **MSA w/ 240** | 90.23       | 93.73       | 64.80        | 82.92     |
> | **MSA w/ 250** | 90.23       | 93.41       | 64.80        | 82.81     |
>
> #### (2) Sensitivity of **Layer Freezing Threshold** $\rho_{\text{layer}}$ (CKA-based)
>
> | $\rho_{\text{layer}}$ | **TIMIT-3** | **SC2**   | **VocalSet** | **ESC-50** | **US8K**  | **Avg.**  |
> | --------------------- | ----------- | --------- | ------------ | ---------- | --------- | --------- |
> | 0.90                  | 83.41       | 90.96     | 62.82        | 94.50      | 97.38     | 85.01     |
> | 0.91                  | 83.41       | 90.96     | 62.82        | 94.50      | 97.38     | 85.01     |
> | 0.92                  | 85.63       | 90.96     | 62.82        | 94.50      | 97.38     | 86.26     |
> | 0.93                  | 85.63       | 90.96     | 62.82        | 94.50      | 97.38     | 86.26     |
> | **0.94**              | **85.63**   | **90.96** | **62.82**    | **94.50**  | **97.38** | **86.26** |
> | 0.95                  | 85.63       | 90.54     | 62.82        | 94.50      | 97.38     | 86.17     |
> | 0.96                  | 85.63       | 90.54     | 62.82        | 92.25      | 97.38     | 85.72     |
> | 0.97                  | 85.63       | 90.54     | 60.23        | 92.25      | 97.38     | 85.21     |
> | 0.98                  | 85.63       | 90.54     | 60.23        | 92.25      | 97.08     | 85.15     |
> | 0.99                  | 85.63       | 90.53     | 60.23        | 92.25      | 97.08     | 85.14     |
> | 1.00                  | 85.63       | 90.53     | 60.23        | 92.25      | 97.08     | 85.14     |
> | w/o                   | 85.63       | 90.53     | 60.23        | 92.25      | 97.08     | 85.14     |
>
> #### (3) Sensitivity of **Boundary Ratio** $\rho_{\text{ratio}}$ (VocalSet)
>
> | Ratio | 0.20  | 0.25  | 0.30  | **0.35**  | 0.40  | 0.45  | 10.50 |
> | ----- | ----- | ----- | ----- | --------- | ----- | ----- | ----- |
> | Acc   | 65.79 | 67.43 | 67.43 | **69.08** | 68.75 | 64.47 | 64.47 |
>
> #### (4) Sensitivity of **Augmentation Strength** (VocalSet)
>
> | Strength | 0.1   | 0.2   | 0.3   | 0.4   | **0.5**   | 0.6   | 0.7   | 0.8   | 0.9   |
> | -------- | ----- | ----- | ----- | ----- | --------- | ----- | ----- | ----- | ----- |
> | Acc      | 65.79 | 66.12 | 64.80 | 63.49 | **69.08** | 68.10 | 65.79 | 64.47 | 61.51 |
>
> PACE achieves robust improvements over a wide range of hyperparameter values. The default values selected in our experiments are not highly sensitive, making PACE **practical and easy to tune** across different scenarios.

---

> ### Author Response · Authors · 2025-11-20
> **Response to Reviewer 4KNg (Part 3)**
>
> ### Q: Clarification on Boundary-Aware Perturbation
>
> Thank you for the insightful comment. We clarify that our **boundary-aware perturbation does not require storing any past backbone**. The confusion likely arose from a brief or ambiguous phrasing in the original manuscript, and we have revised it to avoid potential misinterpretation.
>
> #### No storing Backnone Required
>
> To identify boundary-prone samples at session $t$, we attach a **temporary analytic classifier** to the frozen backbone and fit it solely on the current session's data, **without updating the backbone**. This classifier is used only to identify samples that are misclassified into previously seen classes ($1{:}t-1$) and is immediately discarded after this step.
> Importantly, **the backbone is never duplicated**, i.e., past models are not stored. The procedure incurs no persistent memory overhead beyond holding a lightweight, temporary classifier during sample selection within the current session.
> Once boundary-prone samples are identified, we apply time–frequency perturbations and compute the regularization loss $\mathcal{L}_\text{reg}$, which is combined with the current session's cross-entropy loss to update the backbone. At no point is access to or storage of any previous entire backbone checkpoint required.
>
> #### Why Time–Frequency Masking?
>
> Thank you for the insightful comment. Our use of **time–frequency masking** is grounded in both the spectral structure of audio data and the specific goals of boundary-aware regularization. We have clarified this rationale and included a direct comparison to alternative augmentations in the revised manuscript.
>
> **Why Time–Frequency Masking?**
> We adopt **SpecAugment-style time–frequency masking** [6] because it provides a **label-preserving, online perturbation strategy** that operates in the spectrogram domain. This allows us to construct boundary-prone samples by introducing small, local distortions to energy patterns, without altering the sample's underlying semantics.
> Such perturbations are especially well-suited for boundary-aware regularization, enabling **topology-preserving exploration** near the decision boundary while preserving class identity. This approach directly supports the goal of encouraging intra-class compactness and inter-class separation.
>
> **Why Not Other Augmentations?**
> Alternative audio augmentations often introduce semantic shifts that violate the assumptions required for boundary approximation:
> (1) **Noise-based perturbations**. Noise addition significantly distorts the semantic content. For instance, samples from ESC-50 may drift toward unrelated classes (e.g., engine → rain), no longer lying near the original decision boundary.
> (2) **Pitch shifting**. Particularly in music and speech datasets, pitch conveys core class identity (e.g., musical genre, speaker, or note). Pitch-based augmentation can transform the sample into a different class altogether, invalidating its use for boundary-sensitive regularization.
>
> We **further visualize** the effects of noise-based augmentation and time–frequency masking in $\color{blue} \text{\bf Figs. 8(a), 8(b)}$. As clearly shown, noise-based augmentation causes noticeable semantic drift in a subset of the features.
> In contrast, **time–frequency masking** perturbs low-level spectral cues while preserving high-level semantics, producing class-consistent samples that remain within the natural manifold of each category. This choice directly supports the goals of boundary-aware regularization in a modality-consistent, label-stable manner.
>
> #### Robustness Analysis on Time–Frequency Masking (tf-mask)
>
> To assess whether the proposed regularization term $\mathcal{L}_\text{reg}$ degrades robustness under perturbation, we evaluated models on **VocalSet** with and without **time–frequency masking (tf-mask)** applied at test time:
>
> | Evaluation            | Test w/o tf-mask | Test w/ tf-mask |
> | --------------------- | ---------------- | --------------- |
> | Train **w/o tf-mask** | 66.78            | 60.52           |
> | Train **w/ tf-mask**  | 69.08            | 67.25           |
>
> Models trained with the proposed regularization exhibit **substantially smaller performance drop** under masked inputs (1.83% vs. 6.26%), indicating that $\mathcal{L}_\text{reg}$ actually **enhances robustness**, rather than harming it.
>
> We have added it to our revised manuscript (see $\color{blue} \text{\bf Appendix Section E.7}$).
>
> ---
>
> ### References
>
> [1] LoRA Subtraction for Drift-Resistant Space in Exemplar-Free Continual Learning. CVPR, 2025.
>
> [2] Training networks in null space of feature covariance for continual learning. CVPR, 2021.
>
> [3] SpecAugment: A Simple Data Augmentation Method for Automatic Speech Recognition. Interspeech, 2019.
>
> [4] Panns: Large-scale pretrained audio neural networks for audio pattern recognition. IEEE TASLP, 2020.
>
> [5] Multi-Source Contrastive Learning from Musical Audio. Greeks in AI Symposium, 2025.

---

> > ### Comment · Reviewer_4KNg · 2025-11-24
> > **Response to rebuttal**
> >
> > The authors have effectively addressed my concerns through the detailed rebuttal and the inclusion of new experiments. The restructuring of the paper is also a positive improvement. Given these improvements, I have increased my rating.
> >
> > *Note: Please correct a minor formatting error found in Table 14.

---

> > > ### Author Response · Authors · 2025-11-24
> > >
> > > We have corrected the formatting error in Table 14. Thank you so much for your valuable suggestions and strong support!

---

### Official Review · Reviewer_FS1X · 2025-10-30

**Soundness:** 4
**Presentation:** 4
**Contribution:** 4
**Rating:** 8
**Confidence:** 3

**Summary:**

This paper studies continual learning for audio tasks with pretrained models. It argues that directly using parameter-efficient fine-tuning methods from vision does not work well for audio, since audio backbones focus on low-level spectral cues while continual learning needs high-level semantic adaptation. The authors provide a benchmark and analysis showing two failure modes: representation saturation on coarse-grained tasks and representation shift on fine-grained tasks. They propose PACE, which improves first-session adaptation, applies LoRA only to later layers, uses a subspace-orthogonal strategy for multi-session updates, and adds a boundary-aware regularizer. Experiments across six audio continual learning benchmarks show consistent gains over prior work and a reduced gap to joint training.

**Strengths:**

1. AFAIK, this paper is the first to provide a deep dive into the unique challenges of audio CL with PTMs. The empirical analysis that distinguishes the difficulties in coarse-grained vs. fine-grained audio scenarios (Findings 1, 2, and 3 in Section 2) is a major strength and provides a clear motivation for the proposed method.
2. Their PACE framework is technically sound and its components directly address the problems identified.
3. Their empirical evaluation is thorough. The authors benchmark PACE on six diverse audio datasets, with different granularities and domains. The comparison against a wide range of state-of-the-art CL methods (Table 2) is thorough and convincingly demonstrates the superiority of the method.
4. The paper well written. The figures are effective at conveying their ideas. The structured presentation of findings makes the paper easy to follow and understand.

**Weaknesses:**

We appreciate the detailed analysis and strong empirical results achieved in this submission. To maximize the impact and clarity of the work, we suggest the authors address the following points:

1. The central claim of the paper is about addressing a "fundamental property of audio backbones" in continual learning. Despite this broad claim, all experiments are exclusively conducted using the EAT backbone, which is a spectrogram-based masked prediction model. Demonstrating that the observed challenges (representation shift) and the effectiveness of the PACE solutions hold true for a distinct audio PTM architecture (e.g., BEATs, AST) would significantly strengthen the paper's claims of generality.

2. No explanation on how $\rho_{layer}=0.94$, $\rho_{svd}=0.99$, and $N_{stop}=220$ were chosen. There were multiple continuous variables that were defined without a proper analysis of their impact to the method's performance.

3. The Multi-Session Adaptation (MSA) stage is essential for fine-grained performance. The authors adopt complex strategies, including LoRA Subtraction and subsequent SVD on the uncentered covariance matrix $X^{ucov}_t$, specifically to ensure "computational efficiency" against methods with "extensive storage overheads". However, to fully validate this optimization, a quantitative analysis detailing the increase in memory footprint or training time (e.g., time per session) of the full PACE pipeline compared to simpler, stable FSA baselines (such as RanPAC or ACL) is required.

**Questions:**

1. The stopping threshold for Multi-Session Adaptation (MSA) is determined when the cumulative number of seen samples exceeds $N_{stop} = 220$, which then dictates the final adaptation session $T_3$ (Table 7). $N_{stop}$ appears to be a fixed hyperparameter across all six datasets. Could the authors elaborate on the selection process for the specific value of $220$?  Was this value chosen based on a grid search across datasets?


2. In the improved First-Session Adaptation (Section 3.2), the paper proposes an asymmetric training scheme where the head's learning rate ($\eta_{head}$) is set significantly lower than the backbone's ($\eta_{bb}$). The authors note this is opposite to methods like LAE and SLCA, which suppress backbone updates to mitigate forgetting. Could you specifically elaborate on why this "opposite" approach is beneficial for audio PTMs? I assume it has to do with "compelling the backbone to absorb most gradient signals", which aligns with the finding that adaptation should be restricted to the deeper, semantic layers ($\ell \geq L_{tune}$). Does this suggest that the task-specific semantic information (required by downstream audio tasks) is encoded so poorly in EAT's deeper layers that significant, forced fine-tuning via a high $\eta_{bb}$ is mandatory for effective transfer, unlike in pretrained vision models where freezing often is enough?


3. The boundary-aware regularization component (Section 3.3) uses time-frequency masking ($Q(x_{i,t}, r_T, r_F)$) to generate perturbed samples near decision boundaries (approximating $B_t$). The loss term $\mathcal{L}_{reg}$ is shown to further enhance performance gains provided by MSA (Figs. 6(c) and 6(d)). Given that many audio augmentations exist (e.g., adding noise, pitch shifting), is the choice of time-frequency masking critical to effectively generating the required "boundary-prone samples" $B_t$? A discussion or small ablation comparing time-frequency masking against other standard augmentation techniques for defining $B_t$ would confirm that the choice of perturbation method is specifically aligned with the spectral nature of audio features.

---

> ### Author Response · Authors · 2025-11-20
> **Response to Reviewer FS1X (Part 1)**
>
> Thank you for your thoughtful and constructive feedback. Below we address your comment with a point-to-point response.
>
> ### W1: Additional Backbones
>
> Thank you for the helpful suggestion. We have added the required experiments to our revised manuscript (see $\color{blue} \text{\bf Appendix Section E.5}$). To evaluate whether PACE generalizes beyond a single pretrained model, we conducted new experiments using **SSLAM** [1], a recent source-aware backbone pretrained on polyphonic mixtures.
> We benchmarked PACE against L2P, HiDe-Prompt, and RanPAC on two coarse-grained and two fine-grained datasets. The results are summarized below:
>
> | Dataset  | Granularity    | Backbone | L2P   | HiDe-Prompt | RanPAC | **PACE**  |
> | -------- | -------------- | -------- | ----- | ----------- | ------ | --------- |
> | ESC-50   | Coarse-grained | SSLAM    | 40.50 | 82.00       | 95.75  | **96.25** |
> | SC2      | Coarse-grained | SSLAM    | 15.24 | 37.85       | 88.59  | **90.39** |
> | VocalSet | Fine-grained   | SSLAM    | 17.76 | 47.22       | 63.83  | **68.42** |
> | TIMIT-2  | Fine-grained   | SSLAM    | 0.32  | 46.24       | 90.08  | **93.81** |
>
> These results show that **PACE consistently outperforms all baselines** across both **pretrained checkpoints** and **granularity levels**, highlighting two key points:
> (1) The limitations we diagnose in pretrained audio models (e.g., representation saturation and misalignment) are not specific to EAT and also manifest in more recent, source-aware models like SSLAM.
> (2) PACE generalizes well across different pretrained models, reinforcing its value as a robust CL framework for audio.

---

> ### Author Response · Authors · 2025-11-20
> **Response to Reviewer FS1X (Part 2)**
>
> ### W2 and Q1: Explanation Of How The Continuous Variables Were Selected
>
> Thank you for the valuable suggestion. We have clarified all continuous-variable selections and made these procedures explicit in the revised manuscript (see $\color{blue} \text{\bf Section 4.2}$ and $\color{blue} \text{\bf Appendix Section E.6}$). Below is a detailed breakdown:
>
> **SVD Threshold ($\rho_{\text{svd}}$)**.
> We adopt $\rho_{\text{svd}}$ directly from prior CL works [2, 3]. In our preliminary studies, varying $\rho_{\text{svd}}$ across a broad range (0.90–0.99) yielded nearly identical performance across all datasets. As $\rho_{\text{svd}}$ only determines the numerical rank used for projection (not the learning dynamics), its sensitivity is minimal.
>
> **Stopping Threshold ($N_{\text{stop}}$) and Layer Freezing Threshold ($\rho_{\text{layer}}$)**.
> These two hyperparameters were selected through a joint grid search on fine-grained benchmarks **VocalSet** and **TIMIT-3** where MSA is essential.
>
> - **Select $\rho_{\text{layer}}$**:
>    Using only FSA (no MSA or regularization), we searched over values in ${0.90, 0.92, 0.94, 0.96, 0.98, 1.00}$ and selected the value that best balanced adaptation and stability.
>
> - **Select $N_{\text{stop}}$**:
>   With $\rho_{\text{layer}}$ fixed, we tuned $N_{\text{stop}}$ using MSA without regularization ($L_{\text{reg}}$). As shown below, performance plateaus in the range of 200–250 updates, with $N_{\text{stop}} = 250$ giving strong, robust results. Although this slightly overshoots the transition session $T_3$  for VocalSet (ideal stop at $T_3 = 2$), it still improves over FSA by around 2%, indicating the method's resilience.
>
> **Validation of $\rho_{\text{layer}}$**
> As shown below and $\color{blue} \text{\bf Fig. 6}$, $\rho_{\text{layer}} = 0.94$ avoids premature freezing of adaptable layers (especially in fine-grained tasks like TIMIT-3), while preserving stable low-level features.

---

> > ### Author Response · Authors · 2025-11-20
> > **Response to Reviewer FS1X (Part 3)**
> >
> > In addition to the above analysis, we provide a detailed sensitivity analysis for the remaining key hyperparameters. To improve clarity, we only retain the first and last columns, and highlight only the row indicated in the prompt.
> >
> > #### (1) Sensitivity of **MSA Stopping Threshold** $N_{\text{stop}}$
> >
> > | **Method**     | **TIMIT-2** | **TIMIT-3** | **VocalSet** | **Avg.**  |
> > | -------------- | ----------- | ----------- | ------------ | --------- |
> > | **FSA**        | 85.63       | 89.92       | 62.82        | **79.46** |
> > | **MSA w/ 150** | 88.25       | 92.78       | 62.82        | 81.28     |
> > | **MSA w/ 175** | 89.05       | 93.24       | 66.78        | 83.02     |
> > | **MSA w/ 200** | 88.49       | 93.33       | 66.78        | 82.87     |
> > | **MSA w/ 210** | 89.21       | 93.33       | 66.78        | 83.11     |
> > | **MSA w/ 220** | 89.21       | 93.73       | 66.78        | **83.24** |
> > | **MSA w/ 230** | 90.39       | 93.73       | 66.78        | 83.63     |
> > | **MSA w/ 240** | 90.23       | 93.73       | 64.80        | 82.92     |
> > | **MSA w/ 250** | 90.23       | 93.41       | 64.80        | 82.81     |
> >
> > #### (2) Sensitivity of **Layer Freezing Threshold** $\rho_{\text{layer}}$ (CKA-based)
> >
> > | $\rho_{\text{layer}}$ | **TIMIT-3** | **SC2**   | **VocalSet** | **ESC-50** | **US8K**  | **Avg.**  |
> > | --------------------- | ----------- | --------- | ------------ | ---------- | --------- | --------- |
> > | 0.90                  | 83.41       | 90.96     | 62.82        | 94.50      | 97.38     | 85.01     |
> > | 0.91                  | 83.41       | 90.96     | 62.82        | 94.50      | 97.38     | 85.01     |
> > | 0.92                  | 85.63       | 90.96     | 62.82        | 94.50      | 97.38     | 86.26     |
> > | 0.93                  | 85.63       | 90.96     | 62.82        | 94.50      | 97.38     | 86.26     |
> > | **0.94**              | **85.63**   | **90.96** | **62.82**    | **94.50**  | **97.38** | **86.26** |
> > | 0.95                  | 85.63       | 90.54     | 62.82        | 94.50      | 97.38     | 86.17     |
> > | 0.96                  | 85.63       | 90.54     | 62.82        | 92.25      | 97.38     | 85.72     |
> > | 0.97                  | 85.63       | 90.54     | 60.23        | 92.25      | 97.38     | 85.21     |
> > | 0.98                  | 85.63       | 90.54     | 60.23        | 92.25      | 97.08     | 85.15     |
> > | 0.99                  | 85.63       | 90.53     | 60.23        | 92.25      | 97.08     | 85.14     |
> > | 1.00                  | 85.63       | 90.53     | 60.23        | 92.25      | 97.08     | 85.14     |
> > | w/o                   | 85.63       | 90.53     | 60.23        | 92.25      | 97.08     | 85.14     |
> >
> > #### (3) Sensitivity of **Boundary Ratio** $\rho_{\text{ratio}}$ (VocalSet)
> >
> > | Ratio | 0.20  | 0.25  | 0.30  | **0.35**  | 0.40  | 0.45  | 10.50 |
> > | ----- | ----- | ----- | ----- | --------- | ----- | ----- | ----- |
> > | Acc   | 65.79 | 67.43 | 67.43 | **69.08** | 68.75 | 64.47 | 64.47 |
> >
> > #### (4) Sensitivity of **Augmentation Strength** (VocalSet)
> >
> > | Strength | 0.1   | 0.2   | 0.3   | 0.4   | **0.5**   | 0.6   | 0.7   | 0.8   | 0.9   |
> > | -------- | ----- | ----- | ----- | ----- | --------- | ----- | ----- | ----- | ----- |
> > | Acc      | 65.79 | 66.12 | 64.80 | 63.49 | **69.08** | 68.10 | 65.79 | 64.47 | 61.51 |
> >
> > PACE achieves robust improvements over a wide range of hyperparameter values. The default values selected in our experiments are not highly sensitive, making PACE **practical and easy to tune** across different scenarios.

---

> > > ### Author Response · Authors · 2025-11-20
> > > **Response to Reviewer FS1X (Part 4)**
> > >
> > > ### W3: Quantitative Analysis of Computational Overhead in MSA
> > >
> > > Thank you for pointing this out. We have added it to our revised manuscript (see $\color{blue} \text{\bf Appendix Section E.4}$).
> > >
> > > Here, we clarify the motivation and computational overhead of MSA:
> > >
> > > **First**, the goal of PACE is _not_ to introduce complexity for its own sake. Each component of MSA, including LoRA Subtraction and SVD-based subspace construction, directly follows from our empirical findings in Section 2, which show that pretrained audio backbones suffer from **severe representation shifts** across sessions. These mechanisms form a principled technical baseline grounded in the characteristics of audio continual learning, rather than ad-hoc additions.
> > >
> > > **Second**, regarding computational overhead, MSA _does not_ increase the per-epoch training time inside each session. The gradient projection is a lightweight, closed-form operator applied only to the LoRA gradients, and its backward pass does not add meaningful cost.
> > >
> > > The additional overhead comes from two sources:
> > >
> > > - **Subspace computation between sessions**
> > >    The projection matrix is updated only once after each session and involves an SVD on a small covariance matrix (LoRA rank ≪ backbone dimension).
> > > - **More than one adaptation session**
> > >    Unlike FSA, which adapts only during the first session, PACE may perform a few additional sessions of adaptation on fine-grained datasets. However, thanks to the early stopping rule in MSA, the number of updates is _automatically bounded_ and does not scale linearly with the number of tasks.
> > >
> > > Importantly, for **coarse-grained datasets**, PACE introduces _no_ additional training time beyond RanPAC, because the early-stop mechanism halts adaptation after the first session.
> > >
> > > **Third**, to quantify the overhead, we compare the total training time of PACE with RanPAC (stable FSA baseline) and HiDe-Prompt (a vision-domain PEFT baseline with significantly larger computational burden). The ratios below report total wall-clock training time normalized to RanPAC:
> > >
> > > | Dataset  | Training Time on single NVIDIA-A100 | Training Time Ratio (PACE / RanPAC) | Training Time Ratio (HiDe-Prompt / RanPAC) |
> > > | -------- | ----------------------------------- | ----------------------------------- | ------------------------------------------ |
> > > | Vocalset | 0.22 sec/sample                     | 1.22                                | 5.44                                       |
> > > | TIMIT-3  | 0.12 sec/sample                     | 2.96                                | 146.98                                     |
> > > | TIMIT-2  | 0.31 sec/sample                     | 3.13                                | 124.19                                     |
> > >
> > > These results demonstrate that PACE achieves significant performance gains **with only modest overhead**, while HiDe-Prompt incurs 5× to 40× higher cost compared to PACE despite worse accuracy. Overall, this highlights PACE's strong efficiency–effectiveness trade-off, particularly in fine-grained CL scenarios.

---

> > > > ### Author Response · Authors · 2025-11-20
> > > > **Response to Reviewer FS1X (Part 5)**
> > > >
> > > > ### Q2: Why Use a Lower Learning Rate for the Head and a Higher One for the Backbone?
> > > >
> > > > Thank you for the thoughtful question. As correctly noted, the asymmetric learning rate design is intended to **encourage gradient flow into the backbone** rather than the output classification head, thereby aligning representations with downstream semantics.
> > > > Below, we expand on the motivations and supporting evidence:
> > > >
> > > > **Preventing Representation Saturation**
> > > > Our empirical analysis reveals that on **coarse-grained audio tasks**, pretrained backbones already encode most task-related knowledge, and leads to **representation saturation**. In this situation, the residual task-specific information tends to be absorbed by the output head, not the backbone. This behavior contrasts with vision-domain methods like SLCA or LAE, where the backbone remains the primary locus of adaptation.
> > > >
> > > > Since PACE **replaces the trainable classifier with an analytic head after FSA**, any information stored in a trainable head during adaptation would be discarded. Therefore, we use a **higher learning rate and early freezing strategy for the backbone** to ensure that meaningful gradients are absorbed into the deeper layers of the representation space, where they persist and benefit downstream CL.
> > > >
> > > > **Restricting Adaptation to Later Layers**
> > > > This design is further informed by findings in prior works [4,5], which show that **early layers in pretrained audio models** (e.g., EAT) encode **generic spectral patterns** which serve as essential general knowledge across distributions. Adapting these shallow layers with low-level knowledge to task-specific signals often leads to reduced transferability and increased forgetting.
> > > > Hence, PACE proposes to restrict PEFT updates to the **later layers**, where task-relevant features naturally emerge. The asymmetric learning rate complements this by **amplifying updates where they matter most**, while preserving general-purpose features in earlier layers.
> > > >
> > > > ---
> > > >
> > > > ### Q3: The Choice of Time–Frequency Masking for Boundary-Aware Regularization
> > > >
> > > > Thank you for the insightful comment. Our use of **time–frequency masking** is grounded in both the spectral structure of audio data and the specific goals of boundary-aware regularization. We have clarified this rationale and included a direct comparison to alternative augmentations in the revised manuscript (see $\color{blue} \text{\bf Figs. 8(a), 8(b)}$ and $\color{blue} \text{\bf Appendix Section E.7}$).
> > > >
> > > > **Why Time–Frequency Masking?**
> > > > We adopt **SpecAugment-style time–frequency masking** [6] because it provides a **label-preserving, online perturbation strategy** that operates in the spectrogram domain. This allows us to construct boundary-prone samples by introducing small, local distortions to energy patterns, without altering the sample's underlying semantics.
> > > > Such perturbations are especially suitable for boundary-aware regularization, as they enable **topology-preserving exploration** near the decision boundary while maintaining class identity. This approach directly supports the goal of encouraging intra-class compactness and inter-class separation.
> > > >
> > > > **Why Not Other Augmentations?**
> > > > Alternative audio augmentations often introduce semantic shifts that violate the assumptions required for boundary approximation:
> > > > (1) **Noise-based perturbations**. Noise addition significantly distorts semantic content. For instance, samples from ESC-50 may drift toward unrelated classes (e.g., engine → rain), no longer lying near the original decision boundary.
> > > > (2) **Pitch shifting**. Particularly in music and speech datasets, pitch conveys core class identity (e.g., musical genre, speaker, or note). So similar to noise-based perturbations, pitch-based augmentation may potentially transform the sample into a different class altogether, invalidating its use for boundary-sensitive regularization.
> > > >
> > > > We **further visualize** the effects of noise-based augmentation and time–frequency masking in $\color{blue} \text{\bf Fig. 8(a)}$ and $\color{blue} \text{\bf Fig. 8(b)}$. As clearly shown, noise-based augmentation causes noticeable semantic drift in a subset of the features.
> > > > In contrast, **time–frequency masking** perturbs low-level spectral cues while preserving high-level semantics, producing class-consistent samples that remain within the natural manifold of each category. This choice directly supports the goals of boundary-aware regularization in a modality-consistent, label-stable manner.

---

> > > > > ### Author Response · Authors · 2025-11-20
> > > > > **Response to Reviewer FS1X (Part 6)**
> > > > >
> > > > > ### Reference
> > > > >
> > > > > [1] SSLAM: Enhancing Self-Supervised Models with Audio Mixtures for Polyphonic Soundscapes. ICLR, 2025.
> > > > >
> > > > > [2] LoRA Subtraction for Drift-Resistant Space in Exemplar-Free Continual Learning. CVPR, 2025.
> > > > >
> > > > > [3] Training networks in null space of feature covariance for continual learning. CVPR, 2021.
> > > > >
> > > > > [4] Byol for audio: Self-supervised learning for general-purpose audio representation. IJCNN, 2021.
> > > > >
> > > > > [5] Look, listen, and learn more: Design choices for deep audio embeddings. ICASSP, 2019.
> > > > >
> > > > > [6] SpecAugment: A Simple Data Augmentation Method for Automatic Speech Recognition. Interspeech, 2019.

---

> ### Comment · Reviewer_FS1X · 2025-11-26
>
> I appreciate the authors' effort in addressing the points me and the other reviewers raised and am satisfied with the responses. Happy to mantain my initial positive rating: _8: accept, good paper (poster)_

---

> > ### Author Response · Authors · 2025-11-26
> >
> > Thank you very much for your positive response and valuable suggestions, which have helped us further improve the quality of our manuscript. We look forward to discussing any questions you may have at any time.

---

### Official Review · Reviewer_ViWM · 2025-10-30

**Soundness:** 3
**Presentation:** 2
**Contribution:** 3
**Rating:** 6
**Confidence:** 2

**Summary:**

This paper proposes PACE, an audio continual learning system that combines multiple techniques, including a low learning rate for head learning, Later Layer LoRA for backbone finetuning, Multi-Session Adaptation for continual learning, and Boundary-Aware Perturbation for enlarging the inter-class margin. It shows good results on various benchmarks.

**Strengths:**

1. The experiment results are impressive.
2. Analysis of the challenges of audio continual learning is clear and understandable.
3. The illustrations in the paper are good.

**Weaknesses:**

1. The technical points are scattered:

The techniques presented in this paper are rather fragmented, jumping from one idea to another without a unified narrative. This makes it difficult for readers to clearly understand the motivation behind each technique.

2. Limited novelty of the proposed methods:

The idea of using a lower learning rate for training the head seems more like an empirical tuning strategy rather than a fundamentally new contribution, and the concept of Multi-Session Adaptation is not novel.

3. Insufficient experiments:

The ablation study is not comprehensive enough. For fine-grained datasets, the paper lacks ablation results for each proposed component, and the accuracy V.S. session curves (similar to Figure 2(c)) are also missing.

**Questions:**

Please see the weaknesses.

---

> ### Author Response · Authors · 2025-11-20
> **Response to Reviewer ViWM (Part 1)**
>
> Thank you for your thoughtful and constructive feedback. Below we address your comment with a point-to-point response.
>
> ---
>
> ### W1 and W2: Unified Technical Narrative and Novelty of the Proposed Method
>
> Thank you for raising this important point. We would like to respectfully clarify that the novelty of PACE does not lie in introducing entirely new mechanisms, but rather in **uncovering empirical principles that fundamentally reshape how continual learning (CL) should be approached for pretrained audio models**.
>
> To strengthen this connection, we have reorganized the Design Overview in the method section to ensure that each component of PACE (**improved FSA, MSA, subspace projection, boundary-aware regularization**) is explicitly grounded in the **three key empirical findings** presented in Section 2. We also revised the method section ($\color{blue} \text{\bf Section 3}$) to follow a consistent structure:
> **Finding → Challenge → Design Principle → Technique**, which helps unify the method into a single coherent pipeline rather than presenting isolated modules.
>
> Concretely:
> (1) To our knowledge, this is **the first systematic study of CL with pretrained audio models**, revealing that **audio backbones behave in a fundamentally different manner** from their vision counterparts.
> (2) In the vision domain, recent methods (e.g., SLCA, RanPAC) focus on **where and how to update the pretrained backbone**, and **how to balance and adapt the classification head**. PACE revisits these two aspects in the audio domain and uncovers remarkably distinct behaviors.
> (3) PACE builds upon this diagnosis to revise both aspects, introducing **improved FSA** and **MSA** tailored to pretrained audio models, and further enhancing them with **subspace-orthogonal projection** and **boundary-aware regularization** to mitigate representation shifts across sessions.
>
> While the individual techniques used in PACE may appear lightweight, their integration is **deeply grounded in novel empirical insights**, and they together form a **simple, practical, and principled framework** tailored to the unique challenges of audio CL. Such simplicity is a strength: it makes PACE **interpretable, scalable, and easy to extend**, which we believe is essential for driving progress in this relatively underexplored area.

---

> ### Author Response · Authors · 2025-11-20
> **Response to Reviewer ViWM (Part 2)**
>
> ### W3: Additional Ablation Experiments (Fine-Grained Datasets and Accuracy-vs-Session Curves)
>
> Thank you for this valuable suggestion. In response, we have conducted additional ablations and clarified the results in the revised manuscript (see $\color{blue} \text{\bf Table 4}$ and $\color{blue} \text{\bf Fig. 9}$). Below, we summarize the main findings. Associated figures and tables have also been updated for clarity.
>
> - **Comprehensive ablations on fine-grained datasets**
>
> We now report full ablation results on all fine-grained benchmarks, analyzing the contributions of **First-Session Adaptation (FSA)**, **Gradient Projection (GP)**, and **Boundary-Aware Regularization (Reg)**:
>
> | Method          | VocalSet  | TIMIT-2   | TIMIT-3   |
> | --------------- | --------- | --------- | --------- |
> | w/o FSA         | 61.51     | 75.87     | 75.87     |
> | FSA only        | 62.82     | 85.63     | 89.92     |
> | w/o GP          | 58.55     | 88.01     | 89.05     |
> | w/o Reg Loss    | 66.78     | 89.21     | 93.73     |
> | **PACE (full)** | **69.08** | **90.95** | **94.05** |
>
> These results clearly show that **each component contributes meaningfully to the final performance**.
>
> - **Additional ablations for Multi-Session Adaptation (MSA)**
>
> More detailed MSA ablations appear in **Fig. 6(a),(b)**. We also provide additional case-study visualizations in **Fig. 7** and **Table 5**. We have made them clearer in the revised manuscript.
>
> - **Newly added “Accuracy vs. Session” curves for fine-grained datasets**
>
> We have added Accuracy-vs-Session results for **VocalSet**.
>
>
>
> **PACE (full)**:
> | s1 | s2 | s3 | s4 | s5 | s6 | s7 | s8 |
> |----|----|----|----|----|----|----|----|
> | 100.00 | - | - | – | – | – | – | – |
> | 89.47 | 81.58 | - | – | – | – | – | – |
> | 88.52 | 65.72 | 84.34 | – | – | – | – | -|
> | 88.52 | 60.27 | 70.49 | 76.63 | – | – | – | – |
> | 80.60 | 68.04 | 73.21 | 76.63 | 72.37 | – | – | – |
> | 85.78 | 55.24 | 71.64 | 75.29 | 71.05 | 78.43 | – | – |
> | 84.21 | 50.23 | 70.30 | 71.05 | 71.05 | 66.92 | 100.00 | – |
> | 84.21 | 44.73 | 68.42 | 71.05 | 71.05 | 68.42 | 100.00 | 44.78 |
>
> **PACE w/o MSA**:
> | s1 | s2 | s3 | s4 | s5 | s6 | s7 | s8 |
> |----|----|----|----|----|----|----|----|
> | 100.00 | - | - | – | – | – | – | – |
> | 89.47 | 84.21 | - | – | – | – | – | – |
> | 86.84 | 71.05 | 73.68 | – | – | – | – | – |
> | 86.84 | 71.05 | 55.26 | 73.68 | – | – | – | – |
> | 84.21 | 73.68 | 57.89 | 78.94 | 71.05 | – | – | – |
> | 81.57 | 71.05 | 60.52 | 84.21 | 76.31 | 68.42 | – | – |
> | 79.78 | 65.78 | 55.26 | 76.32 | 63.16 |  65.79 | 97.36 | – |
> | 78.95 | 50.00 | 50.00 | 52.63| 65.79| 60.53 |94.74| 39.47|
>
> **PACE w/o Regularization**:
> | s1 | s2 | s3 | s4 | s5 | s6 | s7 | s8 |
> |----|----|----|----|----|----|----|----|
> | 100.00 | - | - | – | – | – | – | – |
> |84.21 | 78.95 | – | – | – | – | – |-|
> | 78.58 | 65.72 | 79.88 | – | – | – | – | -|
> | 76.17 | 71.68 | 67.33 | 74.20 | – | – | – | – |
> | 76.17 | 68.04 | 57.89 | 73.68 | 67.29 | – | – | – |
> | 72.62 | 59.97 | 62.16 | 71.50 | 66.81 | 83.97 | – | – |
> | 73.68 | 53.31 | 55.26 | 71.05 | 67.29 | 77.42 | 100.00 | – |
> | 73.68 | 50.00 | 60.52 | 65.79 | 65.79 | 71.05 | 97.37 | 50.00 |
>
> **PACE w/o Gradient Projection**:
> | s1 | s2 | s3 | s4 | s5 | s6 | s7 | s8 |
> |----|----|----|----|----|----|----|----|
> | 100.00 | - | - | – | – | – | – | – |
> | 10.12 | 81.58 | – | – | – | – | – | -|
> | 10.52 | 78.53 | 80.19 | – | – | – | – | -|
> | 10.52 | 73.68 | 60.52 | 81.57 | – | – | – | – |
> | 5.26 | 73.68 | 57.89 | 73.68 | 71.05 | – | – | – |
> | 0.00 | 63.15 | 55.26 | 71.05 | 60.52 | 81.57 | – | – |
> | 0.00 | 63.15 | 55.26 | 71.05 | 65.78 | 81.57 | 92.10 | – |
> | 7.89 | 60.52 | 60.52 | 65.79 | 68.42 | 68.41 | 92.11 | 44.74 |
>
> These results clearly show that PACE preserves robust performance across sessions.
> Without **Gradient Projection**, lack of stability, catastrophic forgetting occurs early, despite decent performance on later sessions.
> Without **Regularization**, lack of plasticity, accuracy degrades more gradually, reflecting higher representation overlap between sessions.
> **Full PACE** shows balanced plasticity and stability, maintaining high accuracy throughout.

---

> > ### Author Response · Authors · 2025-11-27
> >
> > Dear Reviewer ViWM,
> >
> > Thank you again for your thoughtful and constructive feedback. We’ve uploaded a detailed rebuttal that addresses your key comments:
> >
> > 1. We clarified the core novelty of PACE as the systematic investigation and empirical insights that drive a unified, principled design (W1, W2).
> >
> > 2. The method section has been reorganized to follow a clear structure: Finding → Challenge → Design Principle → Technique, showing how each component fits into a coherent whole.
> >
> > 3. We added comprehensive ablations on fine-grained datasets and new accuracy-vs-session curves, which confirm the contributions of each design (W3).
> >
> > We hope these updates help address your concerns and merit consideration toward a stronger score. Please don’t hesitate to reach out if you have any further questions.
> >
> > Warm regards,
> >
> > The Authors

---

### Official Review · Reviewer_gWXH · 2025-10-31

**Soundness:** 3
**Presentation:** 3
**Contribution:** 2
**Rating:** 6
**Confidence:** 2

**Summary:**

This work studies continual learning on audio pretrained encoders. The authors argue that continual learning techniques designed for vision do not transfer well to audio and propose **PACE**, a continual learning framework tailored for audio pretrained models.

**Contributions**

1. **Improved first session adaptation.** Freeze early layers and adapt only deeper and more task sensitive layers before moving to analytic inference.
2. **Adaptive subspace orthogonal PEFT for later sessions.** Used to reduce forgetting and follows LoRA subtraction and null space continual learning ideas, so this part is adapted rather than fully novel.
3. **Spectrogram boundary aware perturbations.** Encourage intra class compactness and inter class margin enlargement.
4. **Regularized analytic classifier without rehearsal.** Uses an analytic classifier that avoids storing historical data.

**Strengths:**

- Well written and easy to follow.
- Comparisons are fair and support the claims.
- Addresses area of audio continual learning.

**Weaknesses:**

- **Learning vs forgetting.** Different contributions such as  projection based update that protects past tasks can also limit learning on the current task. Please add an analysis that separates how much performance comes from stability and how much is lost in plasticity.
- **Fine grained tasks are mostly human voices.** Current fine grained results are on speech or voice like data for example TIMIT and VocalSet. A non voice fine grained audio task such as environmental or musical instrument classification would make the claim stronger.
- **Compute and overhead.** PACE introduces extra stages and training which add compute and time requirements compared to RanPAC.
- **Single dataset sessions.** All sessions are constructed from the same dataset. Please add a cross dataset or cross domain setting to show robustness.
- **Only one pretrained Encoder (EAT) is used.** The experiments use only one audio pretrained encoder (EAT). Adding music oriented models such as MERT and speech specific encoders would show that PACE is not tied to a single backbone.

**Questions:**

1. You showed that vision continual learning methods do not transfer well to audio. Do you expect PACE itself to transfer to vision or are parts of it truly audio specific?
2. It was a bit hard to situate PACE among existing audio continual learning works. Is this mainly because there are very few strong audio continual learning baselines?

---

> ### Author Response · Authors · 2025-11-20
> **Response to Reviewer gWXH (Part 1)**
>
> Thank you for your thoughtful and constructive feedback. Below we address your comment with a point-to-point response.
>
> ### W1: Learning vs Forgetting
>
> We appreciate your insightful suggestion to better disentangle the effects of stability and plasticity. To address this, we provide a detailed analysis comparing multi-session adaptation (MSA) with and without gradient projection (GP) on **VocalSet**. The results below are evaluated around the transition session $T_3$ (see Sec. 3.3), where the procedure shifts from backbone adaptation to backbone freezing. We have added it to our revised manuscript (see $\color{blue} \text{\bf Appendix Section E.2}$).
>
> | Method          | Forgetting in ($t\leq T_3$)(Stability) | Ave Max Acc ($t\leq T_3$) (Plasticity) | Ave Acc ($t\leq T_3$) | Ave Acc ($t>T_3$) | Backward Transfer  |
> | --------------- | -------------------------------- | ------------------------------------ | ------------------- | ----------------- | --------- |
> | FSA             | 27.63                            | **92.10**                            | **64.48**           | 60.52             | -10.90    |
> | MSA w/o GP      | 57.90                            | **92.10**                            | 34.21               | 66.67             | -9.02     |
> | **MSA (w/ GP)** | **23.69**                        | 88.16                                | 64.47               | **70.62**         | **-7.14** |
>
> From the table, we observe that **projection-based updating (GP) does not significantly compromise plasticity**: both MSA with and without GP achieve similar average maximum accuracy across sessions $t\leq T_3$. Furthermore, with or without GP, **the representation space continues to evolve positively after the first adaptation session**, as shown by the improved average accuracy in later sessions $t> T_3$ compared to FSA.
> However, **removing GP introduces severe drift in the learned representation space** (see Fig. 3(d)), leading to substantial forgetting of earlier classes—reflected in the high forgetting score (57.90%). This is further corroborated by the backward transfer, which improves from -9.02 to -7.14 when GP is applied. These results demonstrate that projection-based updating effectively mitigates forgetting while preserving the model's adaptability.
>
>
> ---
>
> ### W2 and W4: Additional Datasets on Non-human Voice and Cross-dataset Evaluations
>
> Thank you for the valuable suggestion. We have added new experiments (see $\color{blue} \text{\bf Appendix Section E.3}$) focusing on _fine-grained non-voice audio_ and _cross-domain distribution shifts_, which are central to the scope of our work. Specifically, we introduced: (1) a musical-instrument genre benchmark **GTZAN**, which contains 10 music genres with rich intra-class variation; and (2) a synthetic cross-domain benchmark **ESC-Speech**, which combines samples from ESC-50 (environmental sounds) and SpeechCommands V2 (spoken words).
>
> **Dataset Selection Rationale**:
> In fact, instrument classification tasks (e.g., GTMUSIC) are relatively coarse-grained. In our preliminary experiments, even **a non-music pretrained backbone (EAT) with only improved FSA** achieved **99.8%** accuracy, indicating low distributional complexity and limited relevance to continual learning (CL).
> In contrast, **GTZAN** offers greater intra-class diversity and is thus more suitable for evaluating CL in fine-grained, non-human audio domains.
> Further, **ESC-Speech** creates an intentional domain mismatch (sound vs. speech), simulating realistic deployment conditions involving heterogeneous sources.
>
> For CL, GTZAN uses a 5-session split with 20 samples per class to capture its fine-grained musical variability, while ESC--Speech adopts a 10-session split with 50 samples per class to reflect its cross-domain (sound–speech) shifts.
> We compared PACE against several strong CL baselines:
>
> | Dataset    | Attribute                | L2P   | HiDe-Prompt | RanPAC | **PACE**  |
> | ---------- | ------------------------ | ----- | ----------- | ------ | --------- |
> | GTZAN      | non-human voice          | 10.00 | 51.00       | 73.00  | **78.00** |
> | ESC-Speech | cross-domain (ESC + SC2) | 21.50 | 52.58       | 57.00  | **72.17** |
>
> **Key Findings**:
> **PACE consistently outperforms all baselines** across both settings, demonstrating robustness to intra- and inter-domain shifts.
> In the **music domain**, the representation shift is particularly pronounced, making vision-customized CL methods (e.g., L2P) ineffective.
> These results further validate the benefit of **MSA**, which allows the model to incrementally align with evolving semantic spaces.

---

> ### Author Response · Authors · 2025-11-20
> **Response to Reviewer gWXH (Part 2)**
>
> ### W3: Compute and Overhead
>
> We acknowledge that PACE introduces additional training compared to fully frozen-backbone methods, and we now explicitly discuss this limitation in the revised manuscript (see $\color{blue} \text{\bf Appendix Section E.4}$).
>
> Importantly, **PACE only updates the backbone during the initial few sessions**, until a semantically aligned representation is learned. After this point, the method transitions to **an analytic classifier** and **lightweight subspace-orthogonal updates**, making subsequent sessions highly efficient.
>
> On **coarse-grained datasets**, **improved FSA alone suffices**, as these tasks are already well-aligned with the pretrained backbone. In such cases, PACE achieves near–joint-training performance with **no additional overhead beyond standard fine-tuning**.
>
> On **fine-grained datasets**, PACE remains **substantially more efficient than prompt-based methods** such as HiDe-Prompt. While PACE incurs modest additional cost relative to RanPAC, it avoids the repeated and costly optimization of prompts or LoRA modules performed in HiDe across all sessions.
>
> Below we report the measured training time ratios on representative fine-grained datasets:
>
> | Dataset  | Training Time | Training Time Ratio (PACE / RanPAC) | Training Time Ratio (HiDe-Prompt / RanPAC) |
> | -------- | --------------------------------- |----------------------------------- | ------------------------------------------ |
> | Vocalset | 0.22 sec/sample | 1.22              | 5.44                                       |
> | TIMIT-3  | 0.12 sec/sample | 2.96              | 146.98                                     |
> | TIMIT-2  | 0.31 sec/sample | 3.13              | 124.19                                     |
>
> Notably, we acknowledge that RanPAC with only first-session adaptation incurs almost no training-time overhead, these results demonstrate that PACE achieves significant performance gains **with minor cost per sample**, while HiDe-Prompt incurs 5× to 40× higher cost compared to PACE despite worse accuracy.
>
> ---
>
> ### W5: Additional Backbones
>
> Thank you for the helpful suggestion. We have added it to our revised manuscript (see $\color{blue} \text{\bf Appendix Section E.5}$). To evaluate whether PACE generalizes beyond a single pretrained model, we conducted new experiments using **SSLAM** [1], a recent source-aware backbone pretrained on polyphonic mixtures.
> We benchmarked PACE against L2P, HiDe-Prompt, and RanPAC on two coarse-grained and two fine-grained datasets. The results are summarized below:
>
> | Dataset  | Granularity    | Backbone | L2P     | HiDe-Prompt | RanPAC  | **PACE**    |
> | -------- | -------------- | -------- | ------- | ----------- | ------- | ----------- |
> | ESC-50   | Coarse-grained | SSLAM    |   40.50     | 82.00       | 95.75   | **96.25**   |
> | SC2      | Coarse-grained | SSLAM    |  15.24      | 37.85            | 88.59 | **90.39** |
> | VocalSet | Fine-grained   | SSLAM    |    17.76    | 47.22            | 63.83 | **68.42** |
> | TIMIT-2  | Fine-grained   | SSLAM    |    0.32    | 46.24            | 90.08 | **93.81** |
>
> These results show that **PACE consistently outperforms all baselines** across both **pretrained checkpoints** and **granularity levels**, highlighting two key points:
> (1) The limitations we diagnose in pretrained audio models (e.g., representation saturation and misalignment) are not specific to EAT and also manifest in more recent, source-aware models like SSLAM.
> (2) PACE generalizes well across different pretrained models, reinforcing its value as a robust CL framework for audio.

---

> ### Author Response · Authors · 2025-11-20
> **Response to Reviewer gWXH (Part 3)**
>
> ### Q1: Transferability of PACE to Vision
>
> Thank you for the insightful comment. In short, **PACE is not expected to transfer effectively to the vision domain**, and this is by design.
>
> The primary goal of PACE is to establish an effective and principled **technical baseline for CL in audio**, not as a general-purpose method across modalities. Its design is motivated by the empirical observations presented in **Section 2**, which highlight domain-specific challenges unique to audio CL, particularly: (1) **frequent and pronounced representation shifts** in pretrained audio backbones; and (2) a mismatch between **low-level time–frequency patterns** and the structured semantic alignment needed for CL.
>
> These issues are **not prevalent in current vision CL scenerios**, where pretrained visual features exhibit **much greater stability** across CL sessions (see Fig. 1(a)). As a result, the core motivation behind PACE (i.e., resolving upstream–downstream misalignment and representational shifts) is not directly relevant in regular vision-based CL tasks.
>
> Therefore, we take a **deliberately conservative stance**:
> While it is technically possible to apply PACE to vision tasks, doing so would not be meaningful or interpretable in its current form, as the underlying assumptions do not hold.
> We further expect the insights from our approach to benefit a broader range of scenarios that encounter similar challenges.
>
> ---
>
> ### Q2: Positioning PACE Within Audio CL Works
>
> Thank you for the insightful comment. In short, to our knowledge, **this work is the first to systematically study continual learning (CL) with pretrained models in the audio domain**.
>
> Most prior work on audio CL [1–5] falls into two broad categories:
> (1) **Task-specific scenarios**. These studies often focus on narrow applications such as keyword spotting, underwater sound classification, or specific noise conditions [1, 2]. They are not designed to generalize across diverse audio recognition tasks and often involve constrained datasets or problem settings.
> (2) **Non-pretrained, vision-customized methods**. Other works [3, 4, 5] apply traditional CL strategies (e.g., rehearsal, regularization) to audio, but do so without leveraging pretrained knowledge. These methods typically inherit design choices from the vision domain and do not address the unique representational properties of audio signals or the behavior of audio pretrained models under distribution shift.
>
> In contrast, our work (1) targets **general-purpose audio recognition tasks** using diverse benchmarks; (2) focuses specifically on **pretrained audio models**, which are becoming increasingly central in real-world deployments; and (3) introduces **audio-specific CL insights and methods** that are not simple extensions of vision-customized techniques.
>
> As vision CL research has shifted toward pretrained-model-based paradigms, we believe it is both timely and necessary to develop **audio-native CL grounded in pretrained models**. We hope our work takes an important step in that direction.
>
> ---
>
> ### References
>
> [1] SSLAM: Enhancing Self-Supervised Models with Audio Mixtures for Polyphonic Soundscapes. ICLR, 2025.
>
> [2] Continual Learning for Fake Audio Detection. Interspeech, 2021.
>
> [3] Underwater Acoustic Data Classification Using Continual Learning. IBCAST, 2024.
>
> [4] Few-Shot Continual Learning for Audio Classification. ICASSP, 2021.
>
> [5] Class-Incremental Learning for Sound Event Localization and Detection. ICASSPW, 2025.
>
> [6] Class-Incremental Learning for Multi-Label Audio Classification. ICASSP, 2024.

---

> ### Author Response · Authors · 2025-11-27
>
> Dear Reviewer gWXH,
>
> Thank you for your great efforts and valuable suggestions. We’ve uploaded a detailed rebuttal, which includes:
>
> 1. A plasticity–stability analysis on VocalSet (W1).
>
> 2. New evaluations on non-human voice and cross-domain audio (W2/W4).
>
> 3. A compute cost comparison showing PACE’s efficiency despite strong gains (W3).
>
> 4. Additional experiments using SSLAM to demonstrate backbone generality (W5).
>
> 5. Clarified positioning of PACE as the first audio-native CL method with PTMs (Q2), and its domain-specific design (Q1).
>
> We hope these updates help address your concerns and merit consideration toward a stronger score. Please don’t hesitate to reach out if you have any further questions.
>
> Warm regards,
>
> The Authors

---

### Author Response · Authors · 2025-11-20
**Overall Response**

We thank all reviewers for their constructive and insightful feedback. In the revised maniscript, we have made the following key changes:

- **Clarified the overall narrative of PACE** *(Reviewers `ViWM`, `gWXH`)*
  We have reorganized the method section so that each component (improved FSA, MSA, subspace gradient projection, boundary-aware regularization) maps directly to the three empirical findings outlined in Section 2. This alignment clarifies the progression from empirical diagnosis to method design.

- **Strengthened and expanded experiments** *(Reviewers `gWXH`, `ViWM`, `FS1X`)*
  We have added:
  - new results on fine-grained non-voice and cross-domain audio benchmarks (GTZAN, ESC-Speech),
  - results on an additional pretrained backbone (SSLAM),
  - comprehensive ablation studies and feature visualization (improved FSA, MSA, subspace gradient projection, boundary-aware regularization),
  - accuracy-vs-session curves for fine-grained datasets.

- **Analyzed computational and memory overhead** *(Reviewers `gWXH`, `FS1X`, `4KNg`)*
  We have added a detailed analysis of training time and memory overhead. PACE introduces only modest overhead compared to RanPAC, is significantly more efficient than traditional prompt-based methods like HiDe-Prompt, and crucially does **not** require storing past models for boundary-aware regularization.

- **Added hyperparameter sensitivity analysis** *(Reviewers `4KNg`, `FS1X`)*
  We systematically evaluate the sensitivity of most hyperparameters (MSA stopping threshold, CKA-based layer freezing, boundary ratio, and augmentation strength). The results demonstrate that PACE is robust across a wide range of configurations.

- **Clarified design choices and limitations** *(Reviewers `gWXH`, `FS1X`, `ViWM`)*
  We have elaborated on:
  - the asymmetric learning-rate strategy (backbone vs. output head),
  - the use of time–frequency masking for boundary-aware perturbations,
  - the adaptive stopping criterion in MSA,
  - and we now explicitly discuss limitations and directions for future work.

For ease of review, all revised sections in the manuscript are **highlighted** in $\color{blue}{}\text{blue}$ and **marked with margin notes** that reference the corresponding issue numbers.

---

> ### Author Response · Authors · 2025-11-24
>
> Dear Reviewers,
>
> Thank you again for your thoughtful and encouraging feedback on our submission. We’ve carefully addressed all comments in the rebuttal, including additional experiments and clarifications to further strengthen our contribution.
>
> We would be grateful if you could take a moment to revisit our response. We hope it helps reinforce the strengths of the work. Please contact us if you have any further questions.
>
> Warm regards,
>
> The Authors

---

> > ### Public Comment · ~Harsh_Shukla2 · 2026-07-22
> >
> > Dear Authors,
> >
> > When will the code release?
> >
> > Please mention a possible timeline for the release of code.

---

### Author Response · Authors · 2025-11-29
**Summary of Review Status Prior to Nov 27 Incident**

Dear Area Chair,

We hope this message finds you well. We understand the burden placed on the program committee due to the recent OpenReview bug and sincerely appreciate your efforts in continuing to uphold the integrity of the ICLR review process.

We would like to provide a brief and factual summary of the discussion and review status of our submission **prior to the Nov 27 identity leakage**, for your consideration as the newly assigned area chair.

Our paper received **very positive initial feedback**, with all reviewers indicating a preference to accept. The official scores before the discussion period were:

| Reviewer ID | Initial Score | Post-Rebuttal Score | Notes |
|-------------|---------------|----------------------|--------|
| **gWXH**    | **6**         | —                    | No response posted |
| **ViWM**    | **6**         | —                    | No response posted |
| **FS1X**    | **8**         | **8**                | Confirmed in post-rebuttal comment ($\color{blue}\text{26 Nov 2025, 14:49}$) |
| **4KNg**    | **6**         | **8**                | Confirmed in post-rebuttal comment ($\color{blue}\text{24 Nov 2025, 16:04}$) |

We believe this reflects a strong endorsement of the work, with two responsive reviewers **both explicitly confirming a score of 8** following our detailed rebuttal and manuscript revisions.

If you have any questions or would like additional information, we are happy to provide further details. Thank you again for your time and service to the community.

Warm regards,
The Authors

---

### Meta-Review · Area_Chair_uk3C · 2026-01-08

**Summary:**

The paper investigates the challenges of Continual Learning (CL) in the audio domain when using pretrained models. The reviewers generally praise the thorough empirical analysis that identifies distinct failure modes—representation saturation in coarse-grained tasks and representation shift in fine-grained tasks—which provides a strong motivation for the proposed PACE framework.

This work received good initial scores, and further refinement was conducted during the rebuttal period to address the reviewers' questions.

**Reviewer Concerns:**

See Summary.

**Reviewer Scores:**

See Summary.

---

### Decision · Program_Chairs · 2026-01-26

Accept (Poster)